# Multi-year observations reveal a larger than expected autumn respiration signal across northeast Eurasia

**Brendan Byrne**[1], **Junjie Liu**[1,2], **Yonghong Yi**[3,4], **Abhishek Chatterjee**[1], **Sourish Basu**[5,6], **Rui Cheng**[2],
**Russell Doughty**[2,7], **Frédéric Chevallier**[8], **Kevin W. Bowman**[1,3], **Nicholas C. Parazoo**[1], **David Crisp**[1], **Xing Li**[9],
**Jingfeng Xiao**[10], **Stephen Sitch**[11], **Bertrand Guenet**[12], **Feng Deng**[13], **Matthew S. Johnson**[14], **Sajeev Philip**[15],
**Patrick C. McGuire**[16], and **Charles E. Miller**[1]

[1]Jet Propulsion Laboratory, California Institute of Technology, Pasadena, CA, USA
[2]Division of Geological and Planetary Sciences, California Institute of Technology, Pasadena, CA, USA
[3]Joint Institute for Regional Earth System Science and Engineering, University of California, Los Angeles, CA, USA
[4]College of Surveying and Geo-Informatics, Tongji University, Shanghai, China
[5]Global Modeling and Assimilation Office, NASA Goddard Space Flight Center, Greenbelt, MD, USA
[6]Earth System Science Interdisciplinary Center, University of Maryland, College Park, MD, USA
[7]College of Atmospheric and Geographic Sciences, University of Oklahoma, Norman, OK USA
[8]Laboratoire des Sciences du Climat et de l'Environnement/IPSL, CEA-CNRS-UVSQ,
Université Paris-Saclay, 91191 Gif-sur-Yvette, France
[9]Research Institute of Agriculture and Life Sciences, Seoul National University, Seoul, South Korea
[10]Earth Systems Research Center, Institute for the Study of Earth, Oceans, and Space,
University of New Hampshire, Durham, NH, USA
[11]College of Life and Environmental Sciences, University of Exeter, Exeter EX4 4RJ, UK
[12]Laboratoire de Géologie, Ecole Normale Supérieure/CNRS UMR8538, IPSL, PSL Research University, Paris, France
[13]Department of Physics, University of Toronto, Toronto, Ontario, Canada
[14]Earth Science Division, NASA Ames Research Center, Moffett Field, CA, USA
[15]Centre for Atmospheric Sciences, Indian Institute of Technology Delhi, New Delhi, India
[16]Department of Meteorology and National Centre for Atmospheric Science, University of Reading, Reading, UK

**Correspondence:** Brendan Byrne (brendan.k.byrne@jpl.nasa.gov)

**Abstract.** Site-level observations have shown pervasive cold season $CO_2$ release across Arctic and boreal ecosystems, impacting annual carbon budgets. Still, the seasonality of $CO_2$ emissions are poorly quantified across much of the high latitudes due to the sparse coverage of site-level observations. Space-based observations provide the opportunity to fill some observational gaps for studying these high-latitude ecosystems, particularly across poorly sampled regions of Eurasia. Here, we show that data-driven net ecosystem exchange (NEE) from atmospheric $CO_2$ observations implies strong summer uptake followed by strong autumn release of $CO_2$ over the entire cold northeastern region of Eurasia during the 2015–2019 study period. Combining data-driven NEE with satellite-based estimates of gross primary production (GPP), we show that this seasonality implies less summer heterotrophic respiration ($R_h$) and greater autumn $R_h$ than would be expected given an exponential relationship between respiration and surface temperature. Furthermore, we show that this seasonality of NEE and $R_h$ over northeastern Eurasia is not captured by the TRENDY v8 ensemble of dynamic global vegetation models (DGVMs), which estimate that 47 %–57 % (interquartile range) of annual $R_h$ occurs during August–April, while the data-driven estimates suggest 59 %–76 % of annual $R_h$ occurs over this period. We explain this seasonal shift in $R_h$ by respiration from soils at depth during the zero-curtain period, when sub-surface soils

remain unfrozen up to several months after the surface has frozen. Additional impacts of physical processes related to freeze–thaw dynamics may contribute to the seasonality of $R_h$. This study confirms a significant and spatially extensive early cold season $CO_2$ efflux in the permafrost-rich region of northeast Eurasia and suggests that autumn $R_h$ from subsurface soils in the northern high latitudes is not well captured by current DGVMs.

## 1 Introduction

Boreal and Arctic ecosystems hold vast quantities of soil carbon and play an important role in the global carbon cycle (Schuur et al., 2015). These ecosystems are also experiencing the most rapid climate change (Overland et al., 2018), driving major changes in the carbon cycle, including greening trends (Park et al., 2016), permafrost thaw (Schuur et al., 2015; Turetsky et al., 2019, 2020), and increased fire frequency and intensity (Veraverbeke et al., 2017, 2021). Yet, the impact of these changes on the carbon budget of the region remains uncertain (Schuur et al., 2015; McGuire et al., 2018; Miner et al., 2022). In part, this is due to sparse site-level observations in boreal and Arctic ecosystems, while the limited available observations of high-latitude ecosystems are providing surprises.

A synthesis of Arctic and boreal site-level flux measurements from the literature found pervasive $CO_2$ release during the cold season (Natali et al., 2019) such that the cold season is not a dormant period but strongly impacts annual carbon budgets (Zimov et al., 1993; Björkman et al., 2010; Natali et al., 2019). Particularly strong releases of $CO_2$ have been observed during the early cold season (Commane et al., 2017; Mastepanov et al., 2013; Jeong et al., 2018). This has been linked to the "zero-curtain effect", wherein the air and surface temperatures drop below 0 °C, but deeper soils remain unfrozen for an extended period due to latent heat release (Outcalt et al., 1990; Romanovsky and Osterkamp, 2000; Hinkel et al., 2001; Zona et al., 2016). The result is an "active layer" of unfrozen soil that can persist for months, resulting in greater respiration than would be expected based on air temperature. Both aircraft (Commane et al., 2017) and site-level (Mastepanov et al., 2013; Jeong et al., 2018) measurements have found substantial $CO_2$ release during the zero-curtain period over Alaska (September–December) that is not well captured by our current generation of Earth system models (Commane et al., 2017). Similarly, $CO_2$ mole fraction enhancements within soils have been observed during the zero-curtain period (Wilkman et al., 2021; Raz-Yaseef et al., 2017). Mechanistically, both biological and physical

processes likely contribute to the enhanced early cold season $CO_2$ release. Physically, freezing forces dissolved $CO_2$ out of solution (Bing et al., 2015), which may then be released through mechanical channels and fissures in the soil that form during freezing (Mastepanov et al., 2013; Pirk et al., 2015; Wilkman et al., 2021). Enhanced $CO_2$ effluxes (release to the atmosphere) have also been observed during the spring thaw (Raz-Yaseef et al., 2017; Arndt et al., 2020). This spring signal has been linked to a delayed release of $CO_2$ production from the previous early cold season (Raz-Yaseef et al., 2017), while a rapid warming and introduction of oxygen during snowmelt have also been proposed as contributing to this signal (Arndt et al., 2020). Finally, observed $CO_2$ effluxes during the middle of the cold season (Natali et al., 2019) have been mechanistically linked to microbial respiration that persists at subzero bulk soil temperatures (Rivkina et al., 2000; Panikov et al., 2006; McMahon et al., 2009; Drotz et al., 2010), with a possible additional contribution from the diffusion of stored $CO_2$ that is produced during the non-frozen season (Natali et al., 2019).

Still, the full spatial extent and magnitude of cold season $CO_2$ release are not well characterized due to sparse site-level observations. Here, we employ a "top-down" approach to estimate the seasonal cycle of data-driven carbon fluxes using space-based observations during the period 2015–2019. This approach complements previous site-level analyses by providing $CO_2$ flux constraints on large continental-scale regions. We utilize these data to investigate carbon cycle dynamics over three large regions within Eurasia (Fig. 1), which are defined based on the east–west temperature gradient (see Sect. 2.1), with the coldest region in the east and warmest region in the west. We focus on Eurasia, as much of this region has particularly sparse site-level observations, yet it is experiencing rapid change (Liu et al., 2020; Bastos et al., 2019). We further compare the observationally constrained seasonality of $CO_2$ fluxes to a suite of dynamic global vegetation models (DGVMs) from the TRENDY ensemble (Sitch et al., 2015) version 8 as used in the Global Carbon Budget 2019 (Friedlingstein et al., 2019) (Sect. 3.1). Our study addresses two main questions. (1) Do large-scale observational constraints support enhanced $CO_2$ effluxes during the shoulder seasons at high latitudes? If so, (2) what are the underlying mechanisms driving this behavior?

We first examine the seasonality of net ecosystem exchange (NEE) constrained by atmospheric inversions of retrieved column-averaged dry-air mole fractions of $CO_2$ ($X_{CO_2}$) from the Orbiting Carbon Observatory 2 (OCO-2) (Crisp et al., 2017; Eldering et al., 2017) and by flask and in situ $CO_2$ measurements (Sect. 2.2). Monthly NEE is obtained from version 9 of the OCO-2 Model Inter-comparison Project (v9 OCO-2 MIP) (Peiro et al., 2022). In addition, we perform a set of three higher-temporal-resolution inversions using the CAMS, TM5-4DVar, and CMS-Flux (Carbon Monitoring System Flux) inversion systems to examine

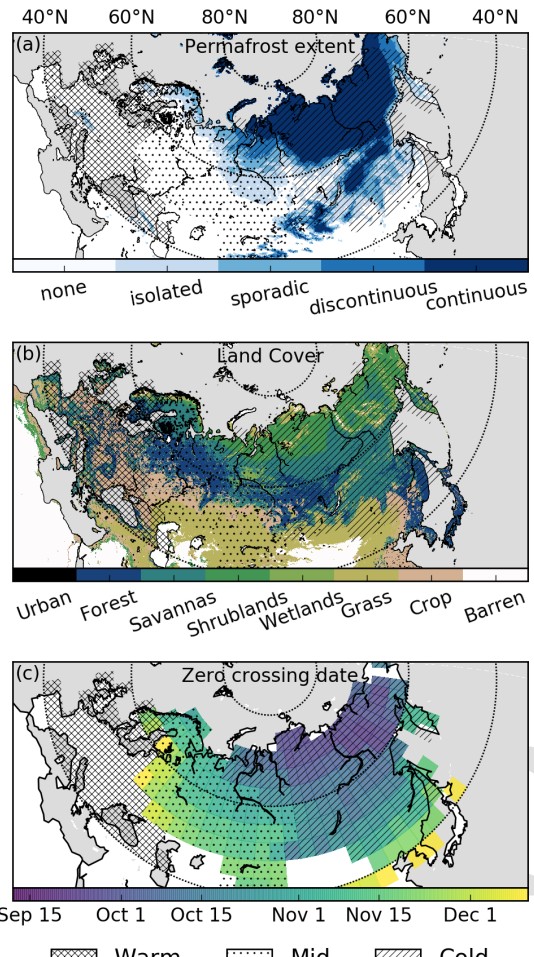

**Figure 1. (a)** Permafrost extent over 2000–2016 (Obu et al., 2018, 2019). **(b)** MODIS International Geosphere Biosphere Programme (IGBP; MOD12C1 v6) land cover for urban areas, forest (tree cover $> 60\%$ and height $> 2\,\mathrm{m}$), savanna (tree cover $10\%$–$60\%$ and height $> 2\,\mathrm{m}$), shrublands (woody perennials cover $> 10\%$ and height $< 2\,\mathrm{m}$), grasslands, croplands, and barren land. **(c)** Zero-crossing date (date when the mean soil temperature drops below $0\,^{\circ}\mathrm{C}$) for the top $0.5\,\mathrm{m}$ of soil from the MERRA-2 Land dataset at $4^{\circ} \times 5^{\circ}$ spatial resolution. Grid cells with no shading do not have a zero-crossing date. Three regions are shown by different hatching patterns. The "Warm" (cross hatching) region does not have a zero-crossing date, the "Mid" (dots) region has a zero-crossing date after 27 October, and the "Cold" (diagonal hatching) region has a zero-crossing date before 27 October. Note that some adjustments from these definitions are made so that the regions are contiguous. The Warm, Mid, and Cold regions have land areas of $5.66 \times 10^{6}\,\mathrm{km^2}$, $8.66 \times 10^{6}\,\mathrm{km^2}$, and $12.65 \times 10^{6}\,\mathrm{km^2}$, respectively.

sub-monthly variability in $CO_2$ fluxes. We then decompose NEE into component fluxes to better understand the processes driving the seasonality of NEE. In particular, we decompose the data-driven NEE fluxes into net primary production (NPP) and heterotrophic respiration ($R_h$):

$$NEE = R_h - NPP. \tag{1}$$

To do this, we use four data-driven gross primary production (GPP) products: FLUXCOM (Jung et al., 2020), FluxSat (Joiner and Yoshida, 2020), the Vegetation Photosynthesis Model (VPM; Zhang et al., 2017), and the Global OCO-2-based solar-induced chlorophyll fluorescence (SIF) product (GOSIF; Li and Xiao, 2019). These datasets use Moderate Resolution Imaging Spectroradiometer (MODIS) reflectances, OCO-2 solar-induced fluorescence, and reanalysis data to infer GPP and thus provide an ensemble of global estimates of GPP to inform its uncertainty. NPP is estimated from GPP using the monthly carbon use efficiency (CUE) from the TRENDY models (Sect. 2.4) using the following relationship:

$$NPP = CUE \times GPP. \tag{2}$$

We then combine the data-driven estimates of NEE and NPP to recover a data-driven seasonal cycle of $R_h$ (Sect. 2.5).

This analysis is performed at two temporal resolutions. First, we leverage the large ensembles from TRENDY and the v9 OCO-2 MIP that provide fluxes at monthly temporal resolution (Sect. 3.1). However, because phenological changes can be significant on shorter timescales (e.g., weekly; Parazoo et al., 2018a), we perform a second analysis at 14 d temporal resolution using three inversion analyses that optimize weekly or 14 d NEE fluxes (Sect. 3.2). For these 14 d fluxes, we further examine mechanistic explanations for data–model differences in $R_h$ using a range of models (Sect. 3.3). Finally, we discuss the results (Sect. 4) and summarize our conclusions (Sect. 5).

## 2 Data and methods

### 2.1 Environmental data and region definitions

We utilize MERRA-2 Land soil temperature data (Reichle et al., 2011, 2017; Gelaro et al., 2017) to define three large regions within Eurasia (Fig. 1). These data were downloaded from the Goddard Earth Sciences Data and Information Services Center at monthly temporal resolution and $4^{\circ} \times 5^{\circ}$ spatial resolution (regridded from model horizontal resolution of $\sim 50\,\mathrm{km}$). Three regions are defined based on the date at which the top $0.5\,\mathrm{m}$ of MERRA-2 Land soil temperature falls below $0\,^{\circ}\mathrm{C}$, referred to as the "zero-crossing date", for a mean seasonal cycle averaged over 4 years (2015, 2016, 2018, and 2019). The "Cold" region has a zero-crossing date before 27 October, the "Mid" region has a zero-crossing date

after 27 October, and the "Warm" region does not have a zero-crossing date. This date was chosen as a cutoff to create two similarly sized Mid and Cold regions. Some adjustments from these definitions are made so that the regions are contiguous. We aggregate the $CO_2$ fluxes described below to these regions by (1) interpolating the Warm, Mid, and Cold regions from $4° \times 5°$ spatial resolution to the grid of the $CO_2$ flux datasets (both GPP and NEE) and (2) calculating the area-weighted net fluxes over the regions. We also obtain the downward shortwave flux from the MERRA-2 Land dataset.

Several datasets are also used for supplementary evaluation of the MERRA-2 Land soil temperature seasonality (Sect. S2 in the Supplement). For that analysis, we use ERA5-Land reanalysis soil temperature data (Muñoz Sabater, 2019), generated using Copernicus Climate Change Service Information 2020. We also examined monthly soil temperature from seven models from the Coupled Model Intercomparison Project Phase 6 (CMIP6) (Eyring et al., 2016) for the historical simulations and Shared Socioeconomic Pathway 585 (ssp585) simulations, which is the highest emission scenario. The CMIP6 simulations were included to compare with MERRA-2 simulated soil temperature over 2010–2019 and to examine possible trends in soil temperature under a high-emission scenario. The model runs are CanESM5 (r1i1p2f1), MIROC ES2L (r1i1p1f2), ACCESS EMS1 (r1i1p1f1), MRI ESM2 0 (historical r1i1p1f1, ssp585 r1i2p1f1), CNRM ESM2 1 (r1i1p1f2), E3SM 1 1 (r1i1p1f1), and UKESM1 0 LL (r4i1p1f2). These models were chosen because they participated in the Coupled Climate–Carbon Cycle Model Intercomparison (C4MIP) (Jones et al., 2016). Finally, we compare the MERRA-2 Land soil temperature to borehole soil temperature measurements over the period 1998–2020, which were downloaded from the Global Terrestrial Network for Permafrost (GTN-P) borehole database (http://gtnpdatabase.org/boreholes, last access: 9 November 2021).

## 2.2 Atmospheric flux inversions

The OCO-2 Model Inter-comparison Project (OCO-2 MIP) provides standardized experimental setups for assimilating atmospheric $CO_2$ to estimate net biosphere exchange (NBE), defined as

$$NBE = NEE + BB, \tag{3}$$

where BB is biomass burning, across a range of inversion systems. The v9 OCO-2 MIP (Peiro et al., 2022) provides ensembles of nine inversion systems that assimilated a standardized set of in situ and flask $CO_2$ measurements for one experiment (referred to as "IS") and OCO-2 ACOS b9 land nadir and land glint $X_{CO_2}$ retrievals for a second experiment (referred to as "LNLG"). We estimate NEE fluxes from v9 OCO-2 MIP NBE fluxes by subtracting biomass burning emission estimates from the Global Fire Emissions Database version 4 (GFED4.1s) (van der Werf et al., 2017). GFED4.1s

provides estimates of biomass burning using MODIS burned area (Giglio et al., 2013), thermal anomalies, and surface reflectance observations (Randerson et al., 2012). Note that biomass burning is a relatively small contribution to NBE over the regions examined here during the study period (2015–2019) (Fig. S1 in the Supplement). The NEE fluxes produced by each ensemble member over northern Eurasia are shown in Fig. S2.

To examine variability in fluxes at the sub-monthly time step, we examine three other inversion NEE estimates that optimize sub-monthly NEE fluxes: TM5-4DVAR$_{14d}$ LNLGIS, CAMS$_{14d}$ LNLGIS, and CMS-Flux$_{14d}$ LNLGIS. These inversions assimilated both in situ and flask $CO_2$ in addition to OCO-2 ACOS b10 land nadir and land glint retrievals. Note that the ACOS b10 retrievals are updated from the b9 retrievals employed in v9 OCO-2 MIP. The prior and posterior NEE fluxes produced by each ensemble member are shown in Fig. S3, and the inversion setups are described below.

TM5-4DVAR is a variational inversion framework based on the TM5 atmospheric tracer transport model (Meirink et al., 2008; Basu et al., 2013). The TM5-4DVAR$_{14d}$ LNLGIS inversion assimilated 10 s averages of OCO-2 ACOS b10 land nadir and land glint measurements concurrently with in situ measurements to optimize weekly NEE and ocean fluxes. The OCO-2 10 s averages were constructed analogous to the b9 10 s averages assimilated by models in v9 OCO-2 MIP (Peiro et al., 2022). The in situ measurements assimilated were updated from Peiro et al. (2022); specifically ObsPack NRT 5.0 was replaced by NRT 5.2. The flux inversion setup was identical to the setup of "TM5-4DVAR" in Peiro et al. (2022), except (i) the inversion was run from 1 June 2014 to 1 February 2021 (instead of 1 September 2014 to 1 June 2019 in Peiro et al., 2022), (ii) ECMWF ERA5 meteorology was used to drive the model instead of ERA Interim, (iii) a $1° \times 1°$ transport grid over North America was nested inside the global $3° \times 2°$ grid to take advantage of the higher in situ data density, and (iv) prior $CO_2$ fluxes were constructed following Weir et al. (2021).

The CAMS$_{14d}$ LNLGIS inversion utilizes the CAMS greenhouse gases inversion system (Chevallier et al., 2005, 2010; Chevallier, 2013) and assimilates OCO-2 land nadir and land glint $X_{CO_2}$ 10 s averages and in situ $CO_2$ measurements concurrently. A variational system is employed to optimize daytime and nighttime NEE at 8 d temporal resolution on a $1.875° \times 3.75°$ model grid. Tracer transport is performed using the Laboratoire de Météorologie Dynamique (LMDz) general circulation model version LMDz6A (Remaud et al., 2018). These data were downloaded from https://atmosphere.copernicus.eu/ (last access: 29 August 2022). Note that CAMS reports NBE – as explained earlier, we estimate NEE using GFED4.1s biomass burning emissions, as was done for the v9 OCO-2 MIP.

The CMS-Flux$_{14d}$ LNLGIS flux inversions are performed using the setup of Byrne et al. (2020b), which uses the

CMS-Flux inversion system that has been developed under the NASA Carbon Monitoring System Flux project (https://cmsflux.jpl.nasa.gov, last access: 29 August 2022) (Henze et al., 2007; Liu et al., 2014). These flux inversions optimize 14 d NEE and ocean fluxes by assimilating OCO-2 ACOS b10 land nadir and land glint "buddy" super-obs concurrently with in situ and flask measurements from version 6.0 of the GlobalView plus package (Masarie et al., 2014; Schuldt et al., 2020). OCO-2 "buddy" super-obs are obtained by averaging individual soundings into super-obs at $0.5° \times 0.5°$ spatial resolution (within the same orbit) with equal weighting following Liu et al. (2017). We assimilate surface-based in situ and flask measurements between 11:00 and 16:00 local time. These data were also pre-filtered to remove observations that were not well simulated by the model (based on a posteriori data–model $\chi^2$ mismatches greater than three for a preliminary flux inversion). For these inversions, ODIAC fossil fuel emissions (Oda and Maksyutov, 2011; Oda et al., 2018) and GFED4.1s biomass burning emissions (van der Werf et al., 2017), including small fires (Randerson et al., 2012), are prescribed but not optimized.

## 2.3 Dynamic global vegetation models (DGVMs)

We use $CO_2$ flux estimates from an ensemble of 15 dynamic global vegetation models (DGVMs) from TRENDY v8 (Sitch et al., 2015). We utilize fluxes simulated by the CABLE-POP, CLASS-CTEM, CLM5.0, DLEM, ISAM, ISBA-CTRIP, JSBACH, JULES, LPJ, LPX-Bern, OCN, ORCHIDEE, ORCHIDEE-CNP, SDGVM, and VISIT DGVMs. We exclude LPJ-GUESS because monthly output on $R_h$ was not available. We utilize monthly GPP, autotrophic respiration ($R_a$), and $R_h$ fluxes from the "S3" experiment that employs time-varying $CO_2$, climate, and land use forcing data. We further calculate NPP from the simulated GPP and $R_a$ data (NPP $=$ GPP $- R_a$) at the models' native resolution. The NEE, NPP, and $R_h$ fluxes produced by each ensemble member are shown in Fig. S4 for the same 3-year period as the data-driven estimates (2015, 2016, and 2018).

We also utilize TRENDY v8 model output to estimate an ensemble of carbon use efficiency (CUE $=$ NPP/GPP) from each DGVM. CUE can become negative during the winter and spring, when GPP is approximately zero but $R_a$ is nonzero. However, we limit CUE values to a range between zero and one. These CUE estimates are then employed to estimate data-driven NPP estimates from the data-driven GPP data (see Sect. 2.4). Figure S5 shows the CUE estimates derived from the TRENDY v8 models.

## 2.4 GPP datasets and NPP estimates

We utilize four data-driven GPP estimates in this analysis: FluxSat, FLUXCOM, VPM, and GOSIF. These datasets differ in inputs and approach.

FluxSat Version 2 (Joiner and Yoshida, 2020) estimates GPP based on Nadir BRDF-Adjusted Reflectance (NBAR) from the MODIS MYD43D product (Schaaf et al., 2002). The GPP estimates are calibrated with the FLUXNET2015 GPP derived from eddy covariance flux measurements at Tier 1 sites (Pastorello et al., 2020).

FLUXCOM upscales $CO_2$ fluxes from flux tower observations using a variety of machine learning methods and forcing datasets (Jung et al., 2020). We examine the ensemble mean of the nine remote sensing (RS) learning algorithms.

VPM is a light use efficiency model that estimates GPP globally using MODIS surface reflectances and NCEP Reanalysis-2 photosynthetically active radiation and temperature data (Xiao et al., 2004; Zhang et al., 2017). The native spatiotemporal resolution of the dataset is 500 m and 8 d. VPM has been shown to agree well with FLUXNET eddy covariance site-level data (Zhang et al., 2017) and with TROPOMI SIF at the global scale (Doughty et al., 2021).

The GOSIF GPP product estimates GPP based on OCO-2 SIF, MODIS EVI, and reanalysis data from MERRA-2 (Li and Xiao, 2019). To generate GPP estimates, first, 8 d globally gridded $0.05° \times 0.05°$ SIF is estimated from the input data using machine learning algorithms. GOSIF GPP is then estimated from the GOSIF SIF estimates using eight SIF–GPP relationships with different forms (universal and biome-specific, with and without intercept). In this analysis we utilize the mean GPP estimate across the eight SIF–GPP estimates.

These four data-driven GPP estimates are shown in Fig. S6. For this analysis, we estimate NPP from these data using the CUE from the TRENDY models. We perform this calculation differently for the monthly analysis and biweekly analysis. For the monthly analysis, we calculate 60 NPP seasonal cycles for each possible combination of the four GPP and 15 CUE seasonal cycles. We then calculate the mean as our best estimate and interquartile range as a metric of uncertainty. For the biweekly analysis, we calculate the best estimate using the mean GPP and CUE seasonal cycles and calculate the uncertainty using the full range of GPP estimates and interquartile range of CUE estimates. This is done differently to match the NEE analysis, which leverages the larger ensemble from the v9 OCO-2 MIP to examine the mean and interquartile spread for the monthly analysis but employs the full range for the smaller biweekly ensemble of three models.

## 2.5 Data-driven $R_h$ estimates

We calculate the seasonal cycle of $R_h$ by combining the data-driven estimates of NPP and NEE using Eq. (1) ($R_h =$ NEE $+$ NPP). We perform this calculation differently for the monthly analysis and biweekly analysis. For the monthly v9 OCO-2 MIP IS- and LNLG-based estimates, we calculate 540 $R_h$ seasonal cycles by combining the 9 data-driven IS or LNLG NEE estimates with the 60 NPP estimates. We then take the mean and interquartile spread as the best estimate

and uncertainty. For the biweekly analysis, we calculate the best estimate of $R_h$ from the best (mean) estimates of NPP and NEE. We then specify the uncertainty to be the full range of $R_h$ estimates calculated from the three biweekly NEE estimates and the NPP range.

## 2.6 Soil carbon decomposition model

We use the soil carbon decomposition model developed in Yi et al. (2015, 2020) to simulate the contribution of soil at different depths to total $R_h$ and NEE fluxes. The soil decomposition model uses multiple litter and soil organic carbon (SOC) pools to characterize the progressive decomposition of fresh litter to more recalcitrant materials, which include three litterfall pools, three SOC pools with relatively fast turnover rates, and a deep SOC pool with slow turnover rates. The litterfall carbon inputs were first allocated to the three litterfall pools depending on the substrate quality of litterfall component and then transferred to the SOC pools through progressive decomposition. We then model the profile of the carbon pools through accounting for the vertical carbon transport (Yi et al., 2020). A constant diffusivity rate was assigned to permafrost ($5.0\,\mathrm{cm^2\,yr^{-1}}$) and nonpermafrost ($2.0\,\mathrm{cm^2\,yr^{-1}}$) regions within the top 1 m soil and then linearly decreased to 0 at the 3 m b.s. (below surface) (Koven et al., 2013). The boundary conditions at the soil surface were set as the carbon input rate to the three surface litterfall pools. A zero-flux was assigned at the bottom of the soil carbon pool, which was set as 3 m. This accounts for the upper permafrost layer, while carbon in deeper layers (e.g., 3–10 m) is largely insulated from climate variability and ignored in this study. The decomposition rate ($\mathrm{d^{-1}}$) for each carbon pool was derived as the product of a theoretical maximum rate constant and dimensionless multipliers for soil temperature and liquid water content constraints (Yi et al., 2015). In this study, the decomposition was driven by the MERRA2 soil temperature data. For simplicity, the soil saturation was assumed as 1.0 when soil temperature is above $0\,^\circ\mathrm{C}$, while the maximum liquid soil water fraction was used for below freezing (Schaefer and Jafarov, 2016).

## 2.7 FLUXNET data and processing

We examine 15 high-latitude FLUXNET2015 sites to confirm the seasonality of carbon fluxes inferred from the atmospheric $CO_2$ and remote sensing datasets. These sites are listed in Table S1. For this, we utilize monthly data with the quality flag greater than 0.75. We calculate NPP and $R_h$ for each site from the NEE and GPP datasets by applying the CUE from the TRENDY DGVMs at the grid cell containing the FLUXNET site.

# 3 Results

## 3.1 Differences between data-driven and DGVM carbon fluxes

We first examine the mean seasonal cycle of monthly NEE from the v9 OCO-2 MIP inversions and TRENDY v8 DGVMs over the three northern Eurasian regions (mean over 2015, 2016, and 2018; 2017 is excluded due to an OCO-2 data gap during August–September). The objective of this initial analysis is to identify the seasonal features of NEE over northern Eurasia and identify how the data-driven and simulated estimates differ. The spread among the TRENDY v8 models is large and encompasses the data-driven estimates (Fig. S4). Thus, to identify data–model differences, we focus on differences in the ensemble mean estimates and adopt the interquartile spread across v9 OCO-2 MIP and TRENDY to quantify uncertainty in this estimate.

Figure 2a–c show the NEE fluxes for the v9 OCO-2 MIP and TRENDY DGVMs for three regions over Eurasia. The two v9 OCO-2 MIP ensembles (IS and LNLG) generally show close agreement and coherent differences from the TRENDY models (and prior NEE estimates; Fig. S2). The largest differences between the IS and LNLG ensembles occur over the Cold region, where the IS ensemble suggests increased uptake during July and somewhat increased release during October. Still, the coherent differences between the data-driven fluxes (both IS and LNLG) relative to the TRENDY ensemble gives us increased confidence that these inversions are precisely capturing the seasonality of NEE. The comparatively good agreement between the IS and LNLG inversions (relative to TRENDY) also suggests that artifacts related to observational coverage (Byrne et al., 2017; Basu et al., 2018) and data and model biases (Schuh et al., 2019) do not strongly impact the results, although we note that both datasets have spatial and seasonal gaps over northern Eurasia (e.g., Figs. S7–S8 TS1) as discussed in Sect. 4.2. The accuracy of the v9 OCO-2 MIP fluxes is supported through an evaluation of the posterior $CO_2$ fields against independent atmospheric $CO_2$ measurements by Peiro et al. (2022), as well as a supplementary comparisons of the CMS-Flux$_{14d}$ inversions with aircraft data over Alaska (Sect. S1, Figs. S9–S10 TS2).

Comparing the v9 OCO-2 MIP and TRENDY NEE estimates, good agreement is found for the Warm and Mid regions, while larger differences are found for the Cold region. In the Warm and Mid regions, systematic differences exceed the interquartile range during September–October, when the TRENDY models suggest a weaker efflux of $CO_2$ to the atmosphere ($0.30$–$0.51\,\mathrm{g\,C\,m^{-2}\,d^{-1}}$). The TRENDY models also tend to show weaker uptake by land during June–July in the Mid region ($0.05$–$0.35\,\mathrm{g\,C\,m^{-2}\,d^{-1}}$). For the Cold region, the TRENDY models produce weaker carbon uptake during June–July ($0.48$–$0.83\,\mathrm{g\,C\,m^{-2}\,d^{-1}}$) but stronger uptake (or reduced efflux) during August–October ($0.31$–

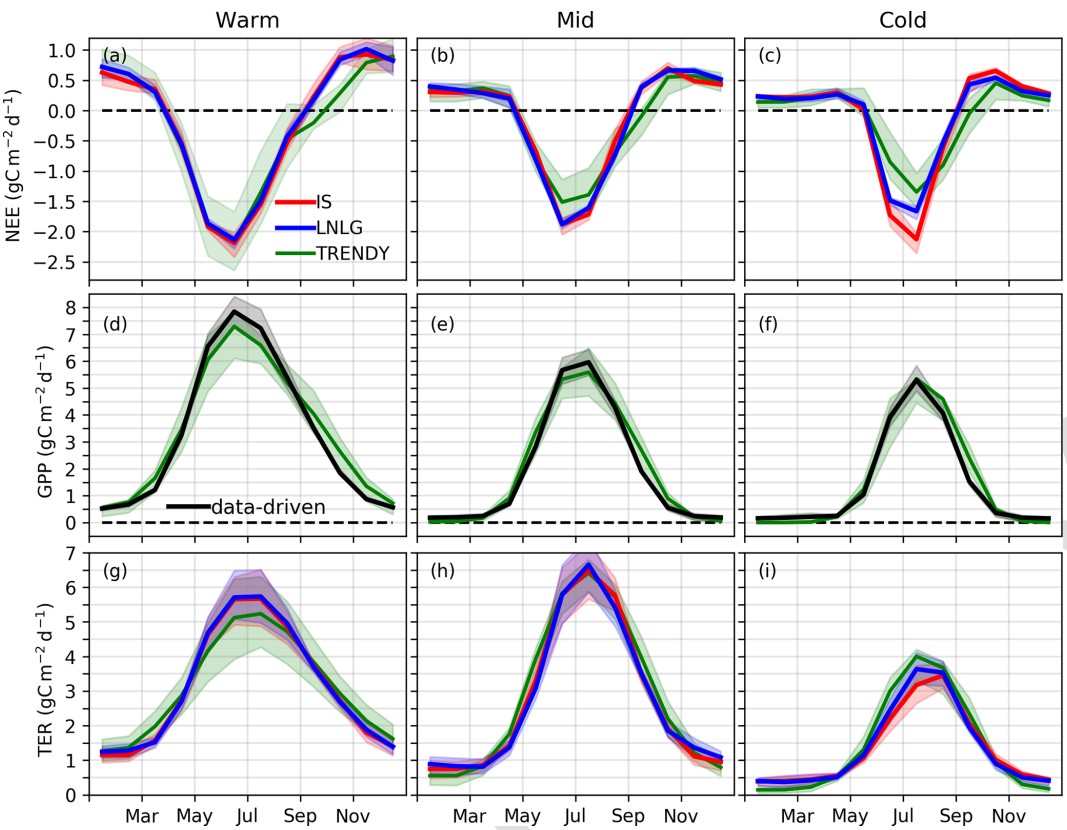

**Figure 2.** Monthly carbon cycle fluxes (average of 2015, 2016, and 2018; 2017 is excluded due to an OCO-2 data gap). **(a–c)** Mean (solid line) and interquartile range (shaded area) of NEE for the ensemble of IS (red) and LNLG (blue) v9 OCO-2 MIP and for the TRENDY ensemble (green). **(d–f)** GPP for the TRENDY ensemble (green) and data-driven datasets (black). **(g–i)** TER simulated by the TRENDY ensemble (green) and calculated from combining the data-driven GPP with the IS (red) and LNLG (blue) v9 OCO-2 MIP NEE constraints.

$0.36\,\mathrm{g\,C\,m^{-2}\,d^{-1}}$). This large amplitude of the NEE seasonal cycle contributes to the large seasonality in $X_{\mathrm{CO_2}}$ observed over eastern Eurasia (Jacobs et al., 2021).

To further investigate the causes of differences in NEE between the TRENDY and v9 OCO-2 MIP ensembles, we separately examine component primary productivity and respiration fluxes. For the most direct decomposition, we employ the data-driven GPP estimates to decompose NEE into GPP and terrestrial ecosystem respiration (TER) fluxes (Fig. 2). This comparison shows that the TRENDY ensemble mean GPP tends to overestimate the data-driven GPP during the autumn (September–November), largely explaining the mismatch in NEE during this season. For TER, we find good agreement for over the Warm region except for an underestimate of TER for the TRENDY ensemble mean during the summer (mirroring GPP). For the Mid region, agreement is found between the TRENDY and data-driven TER estimates throughout the growing season. For the Cold region, we find that the TRENDY ensemble mean suggested greater TER during May–August, which drives the mismatch found in NEE.

We next decompose NEE into component NPP and $R_{\mathrm{h}}$ fluxes. These estimates require an additional assumption about the CUE in comparison to the GPP/TER decomposition but also have the potential to provide more process understanding. As described in Sect. 2.4, we employ the monthly CUE estimates from the ensemble of TRENDY models. This both allows an "apples-to-apples" comparison with the TRENDY models as the CUE estimates are consistent between the data-driven and TRENDY estimates and allows us to propagate uncertainty in CUE from the ensemble spread. The data-driven NPP and TRENDY NPP are shown in Fig. 3d–f. The seasonality in NPP between the data-driven and TRENDY estimates show good agreement for all regions. In the Mid and Warm regions, the TRENDY model mean NPP tends to be lower than the data-driven estimates during June–July ($0.26$–$0.37\,\mathrm{g\,C\,m^{-2}\,d^{-1}}$). However, the largest differences are for the Cold region, where the TRENDY ensemble mean shows increased NPP during August–September ($0.38\,\mathrm{g\,C\,m^{-2}\,d^{-1}}$). This largely accounts for the lower NEE during August–September (86 %–95 %). Thus, despite previously reported deficiencies in model representation of photosynthesis over high latitudes

(Rogers et al., 2017, 2019), we find that TRENDY NPP estimates largely capture the data-driven seasonality and do not drive NEE differences against the data-driven seasonal cycle.

Finally, we compare TRENDY $R_h$ to data-driven $R_h$ (Fig. 3g–i). In the Warm region, the TRENDY model mean $R_h$ is lower than the data-driven estimates during May– September ($0.21–0.26\,\mathrm{g\,C\,m^{-2}\,d^{-1}}$), but the seasonality is similar. In the Mid region, the data-driven and TRENDY $R_h$ seasonal cycles show good agreement throughout the growing season. The largest differences between data-driven and TRENDY $R_h$ seasonal cycles are found for the Cold region. The TRENDY model mean shows increased $R_h$ during May– July ($0.32–0.58\,\mathrm{g\,C\,m^{-2}\,d^{-1}}$) but show reduced $R_h$ during the rest of the year ($0.29–0.51\,\mathrm{g\,C\,m^{-2}\,d^{-1}}$). As a result, the seasonality of data-driven $R_h$ is shifted later in the year relative to the TRENDY ensemble. This can be seen in the cumulative fraction of annual $R_h$, which quantifies the fraction of total $R_h$ released as the season progresses (Fig. 3j–l). The percentage of total annual $R_h$ released during May–July is 46 % for the TRENDY ensemble mean but 37 % (30 %) for the LNLG (IS) data-driven $R_h$ ensemble mean.

We independently confirm a shift in the seasonality of data-driven $R_h$ relative to TRENDY for 15 high-latitude FLUXNET sites (Fig. S11 TS3). Due to the sparsity of FLUXNET sites over northeastern Eurasia, we include sites outside of the "Cold" domain but that have early zero-crossing dates (estimated by a mean October air temperature less than $2\,^\circ\mathrm{C}$). The observed mean $R_h$ peaks across these sites during September, in agreement with the data-driven $R_h$ seasonality. In contrast, the TRENDY mean $R_h$ peak occurs during July (consistent with the regional-scale analysis). This phase shift is also evident in the cumulative fraction of annual $R_h$, which shows that the percentage of total annual $R_h$ released during May–July is 46 % for the TRENDY ensemble mean but 35 % for the FLUXNET-based ensemble mean.

Overall, these results indicate good agreement between the TRENDY ensemble and data-driven estimates for the Warm and Mid regions but show marked differences over the Cold region. In particular, we find that the data-driven estimates suggest a seasonal redistribution of $R_h$ with a reduction during May–July but an increase for the remainder of the year. Further, these results show that differences in $R_h$ largely account for the differences between the data-driven and TRENDY NEE fluxes over the Cold region, except in August–September when NPP differences are large. In the remaining sections, we will characterize the data-driven seasonal cycle of NEE, NPP, and $R_h$ at a higher (biweekly) temporal resolution and investigate mechanistic explanations for the data–model differences found over the Cold region.

## 3.2   Data-driven biweekly CO$_2$ fluxes

We now investigate the data-driven seasonal cycle of NEE, NPP, and $R_h$ with biweekly (14 d) temporal resolution. This higher resolution better resolves temporal changes in CO$_2$

fluxes throughout the growing season, particularly during the shoulder seasons, when week-to-week changes in CO$_2$ fluxes are large (Parazoo et al., 2018a). For this analysis, we utilize a set of three flux inversions that assimilate both in situ and OCO-2 land nadir and glint data (ACOS v10) to estimate sub-monthly CO$_2$ fluxes (TM5-4DVAR$_{14d}$, CAMS$_{14d}$, and CMS-Flux$_{14d}$; individual model fluxes shown in Fig. S3). These inversions give similar NEE seasonality to the v9 OCO-2 MIP monthly fluxes (e.g., Fig. S9 TS4) and have seasonality similar to the v9 OCO-2 MIP LNLG inversions for the Cold region. For NPP, we utilize the same datasets as Sect. 3.1 but at 14 d temporal resolution. We examine the ensemble means for a best estimate and take the full range of model estimates as an illustration of the uncertainty.

Figure 4 shows the 4-year-mean (2015, 2016, 2018, and 2019; 2017 is excluded due to an OCO-2 data gap in summer) seasonal cycle of NEE, NPP, and $R_h$ for the three regions of Eurasia. NEE largely tracks the inverted seasonality of NPP, although peak NPP is slightly delayed relative to peak drawdown in NEE (by 0–2 weeks). Both NEE and NPP generally follow the seasonal cycle of insolation but are somewhat delayed in the Mid and Cold regions likely due to temperature limitation (Liu et al., 2020). Peak $R_h$ is found to be delayed relative to peak NPP by 0–8 weeks. For the Warm and Mid regions, $R_h$ follows the seasonal cycle of surface temperature, with 48 % and 51 % of the annual total $R_h$ occurring after the peak in surface temperature, respectively. In contrast, the Cold region shows a substantial delay relative to surface temperature, with 63 % of the total $R_h$ occurring after the peak in surface temperature. The mean $R_h$ seasonal cycle is also found to have a double peak in this Cold region: a smaller peak of $0.77\,\mathrm{g\,C\,m^{-2}\,d^{-1}}$ occurs during late May followed by a larger peak of $1.70\,\mathrm{g\,C\,m^{-2}\,d^{-1}}$ at the beginning of September. This May peak roughly aligns with the spring thaw and positive zero-crossing at the beginning of May. A potential mechanistic explanation for a spring pulse of $R_h$ could be due to thawing soils that release CO$_2$ that had been trapped within subsurface soil layers over the winter (see Sect. 4). Another plausible mechanism could be the timing of snowmelt, which may insulate the soil over winter (Yu et al., 2016). However, the signal from this first peak is small relative to the uncertainties.

## 3.3   Mechanistic drivers of late-season $R_h$

Data-driven $R_h$ for the Cold region indicates a delayed peak relative to surface temperature and the TRENDY model mean. Here we examine possible mechanistic explanations for this late season peak in $R_h$ using models. We investigate two factors that could potentially contribute to the delay in $R_h$. (1) The first factor is seasonal variations in the labile carbon pool. Leaf and fine root litter carbon pools tend to increase over the growing season as carbon is sequestered through photosynthesis (Randerson et al., 1996). Thus, increased substrate availability in the autumn relative to the

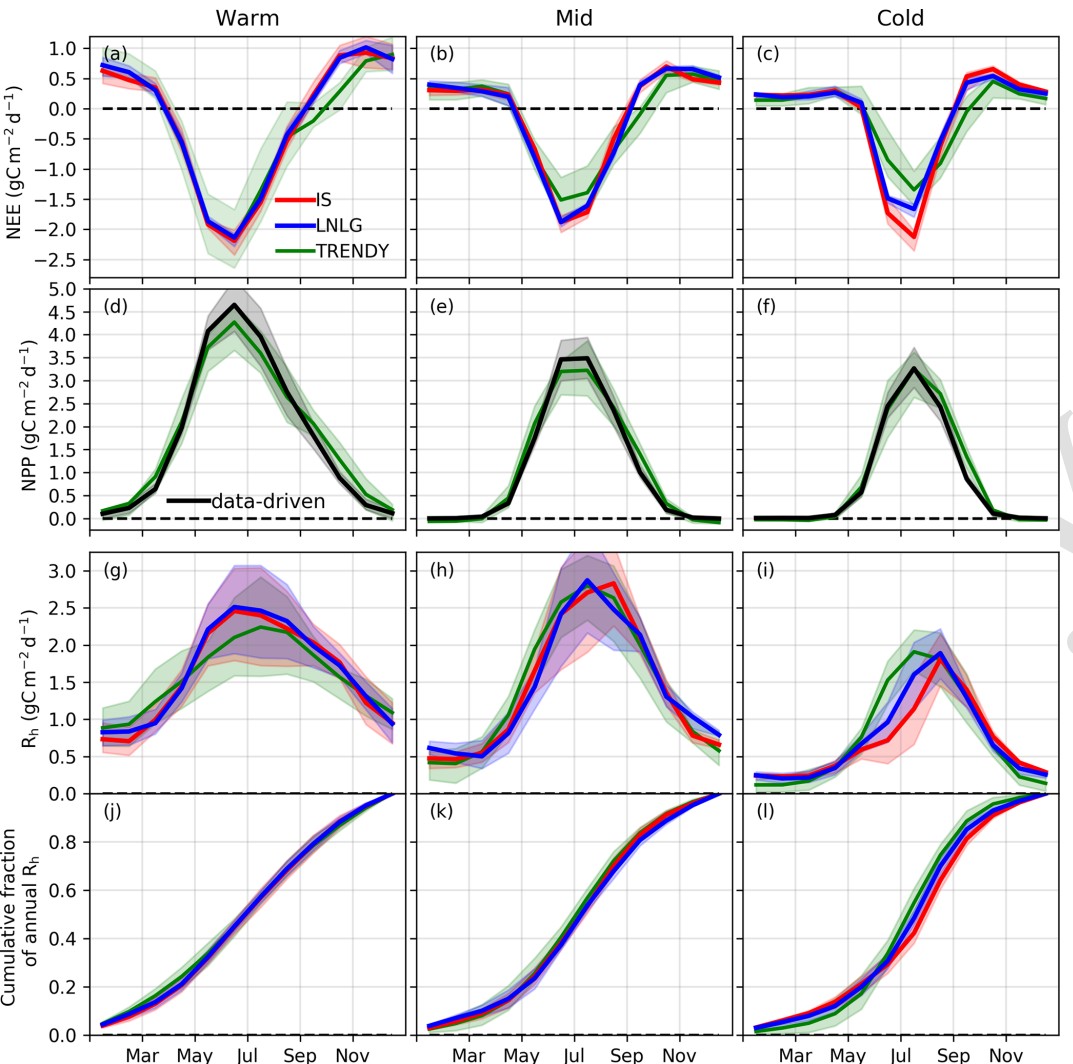

**Figure 3.** Monthly carbon cycle fluxes (average of 2015, 2016, and 2018; 2017 is excluded due to an OCO-2 data gap). **(a–c)** Mean (solid line) and interquartile range (shaded area) of NEE for the ensemble of IS (red) and LNLG (blue) v9 OCO-2 MIP and for the TRENDY ensemble (green). **(d–f)** NPP for the TRENDY ensemble (green) and estimated from data-driven GPP (black). **(g–i)** $R_h$ simulated by the TRENDY ensemble (green) and calculated from combining the data-driven NPP with the IS (red) and LNLG (blue) v9 OCO-2 MIP NEE constraints. **(j–l)** Cumulative fraction of $R_h$ over the growing season.

spring will act to shift the seasonal cycle of $R_h$ later in the year. (2) The second factor is $R_h$ from subsurface soil layers that have a delayed seasonal cycle driven by a lag in soil temperature. Heating and cooling at the surface slowly diffuses through the soil column resulting in a lagged seasonal cycle of temperature with depth (Parazoo et al., 2018b). Figure 5a–c show the seasonal cycle in soil temperature from the MERRA-2 Land dataset. The phase shift in soil temperature seasonality with depth can be up to several months and is largest for the colder regions. Note that we verify the fidelity of the MERRA-2 Land soil temperature against borehole measurements and against simulated soil temperature from ERA-5 reanalysis and the CMIP6 models (see Sect. S2, Figs. S12–S13 TS6).

To test the impact of these factors, we consider a single layer model that represents $R_h$ using a exponential relationship with temperature:

$$R_h = \alpha e^{\beta T}, \tag{4}$$

where $\alpha$ represents the labile carbon pool size, $\beta$ is a constant, and $T$ is the temperature of the carbon pool. To investigate the impact of seasonal and vertical variations in labile carbon, we consider three cases:

1. $R_h(\alpha_c, T_{surf}) = \alpha_c e^{\beta T_{surf}}$: the carbon pool is constant in time ($\alpha_c$), and the surface temperature ($T_{surf}$) drives $R_h$;

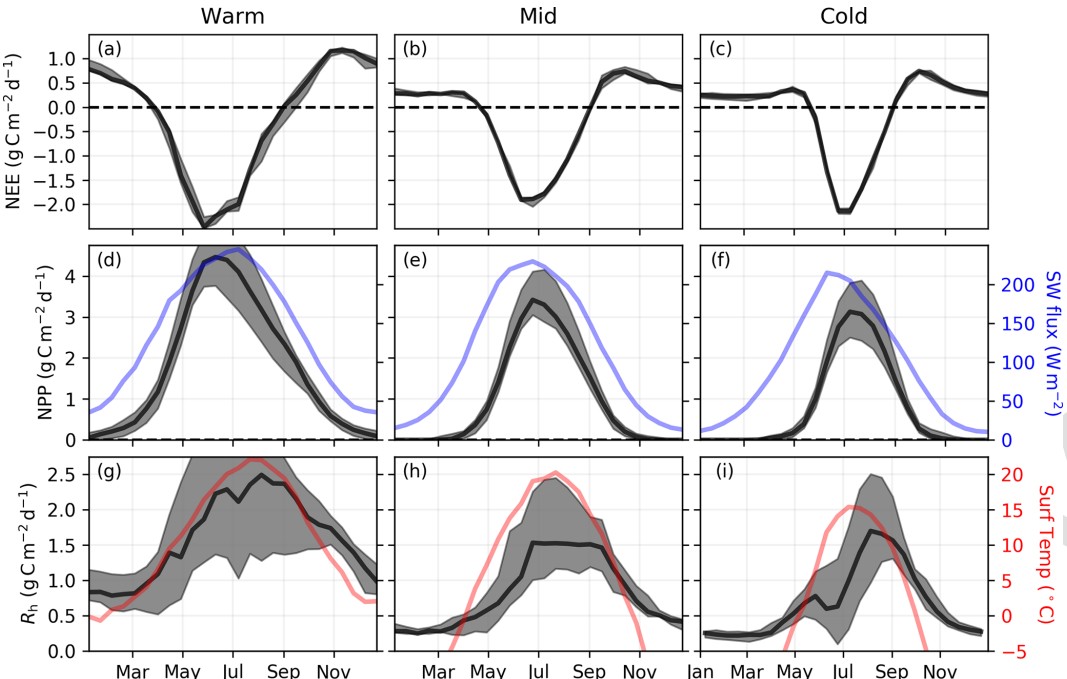

**Figure 4.** TS5 Data-driven 4-year-mean (2015, 2016, 2018, and 2019) 14 d NEE, NPP, and $R_h$. **(a–c)** Mean and ensemble spread of NEE for the CMS-Flux$_{14d}$, TM4-4DVar$_{14d}$, and CAMS$_{14d}$ flux inversions. **(d–f)** Mean and ensemble spread for the data-driven NPP. MERRA-2 Land net downward shortwave flux over land is shown in blue. **(g–i)** Ensemble estimate of $R_h = \text{NEE} + \text{NPP}$ estimated from the three NEE and NPP estimates. MERRA-2 Land surface temperature is shown in red.

2. $R_h(\alpha_c, T_{1\,m}) = \alpha_c e^{\beta T_{1\,m}}$: the carbon pool is constant in time ($\alpha_c$), and the average top meter soil temperature ($T_{1\,m}$) drives $R_h$;

3. $R_{h(\alpha_t, T_{surf})} = \alpha_t e^{\beta T_{surf}}$: the carbon pool is dynamic in time ($\alpha_t = f(t)$, described in the Appendix), and the $T_{surf}$ drives $R_h$. We assume seasonal variations in the carbon pool are within $\pm 15\%$ of the mean ($\gamma = 0.15$, Fig. S14 TS8 shows the seasonal variation in the labile carbon pool).

Figure 5d–f show linear regressions for each one-layer model against the mean biweekly estimate of $R_h$. In each case, the parameters $\alpha$ and $\beta$ are optimized (note linear regressions are performed on $\ln(R_h) = \ln(\alpha) + \beta T$). Statistics on the model fits are provided in Table 1. For the Warm region, all models are able to fit the data well ($R^2 = 0.89$–$0.93$, SE $= 0.051$–$0.067\,\text{g C m}^{-2}\,\text{d}^{-1}$). Similarly, all models are able to largely capture the seasonal cycle in the Mid region ($R^2 = 0.84$–$0.94$), although $R_h(\alpha_c, T_{1\,m})$ appears to better capture the shoulder seasons and gives a smaller standard error (SE $= 0.046\,\text{g C m}^{-2}\,\text{d}^{-1}$) than the other models (SE $= 0.074$–$0.091\,\text{g C m}^{-2}\,\text{d}^{-1}$). The models diverge the most for the Cold region. $R_h(\alpha_c, T_{surf})$ gives the poorest performance ($R^2 = 0.66$, SE $= 0.16\,\text{g C m}^{-2}\,\text{d}^{-1}$), as the driving temperature data peak too early to capture the seasonality of $R_h$. $R_h(\alpha_t, T_{surf})$ performs somewhat better, as the peak in model $R_h$ is delayed relative to surface temperature

($R^2 = 0.77$, SE $= 0.13\,\text{g C m}^{-2}\,\text{d}^{-1}$). Still, $R_h(\alpha_c, T_{1\,m})$ performs the best ($R^2 = 0.88$, SE $= 0.08\,\text{g C m}^{-2}\,\text{d}^{-1}$) and best captures the delayed $R_h$ seasonality relative to surface temperature.

To further confirm that $R_h(\alpha_c, T_{1\,m})$ best captures the seasonality of $R_h$, we fit these same models to seasonal FLUXNET $R_h$ averaged over cold sites. This is a rather rough comparison as we drive the models with soil temperatures averaged over the Cold region rather than site specific datasets (due to absence of soil temperature data). Figure S15 TS10 shows the resulting fits, and Table S2 gives the statistics of the fits. We find that $R_h(\alpha_c, T_{1\,m})$ performs best ($R^2 = 0.96$, SE $= 0.08\,\text{g C m}^{-2}\,\text{d}^{-1}$), while $R_h(\alpha_t, T_{surf})$ performs second best ($R^2 = 0.85$, SE $= 0.19\,\text{g C m}^{-2}\,\text{d}^{-1}$), and $R_h(\alpha_c, T_{surf})$ gives the poorest performance ($R^2 = 0.75$, SE $= 0.25\,\text{g C m}^{-2}\,\text{d}^{-1}$), consistent with the regional-scale data-driven analysis.

This analysis demonstrates that the seasonality of $R_h$ in the Warm and Mid regions is reasonably explained by seasonal variations in $T_{surf}$ but that inclusion of seasonal variations in the labile carbon pool and the impact of soil temperature with depth still improve the seasonal fit. However, for the Cold region, the seasonality of $R_h$ is not well captured by $T_{surf}$, and additional factors, particularly the impact soil temperature with depth, are required to explain the delayed seasonality of $R_h$ over the Cold region.

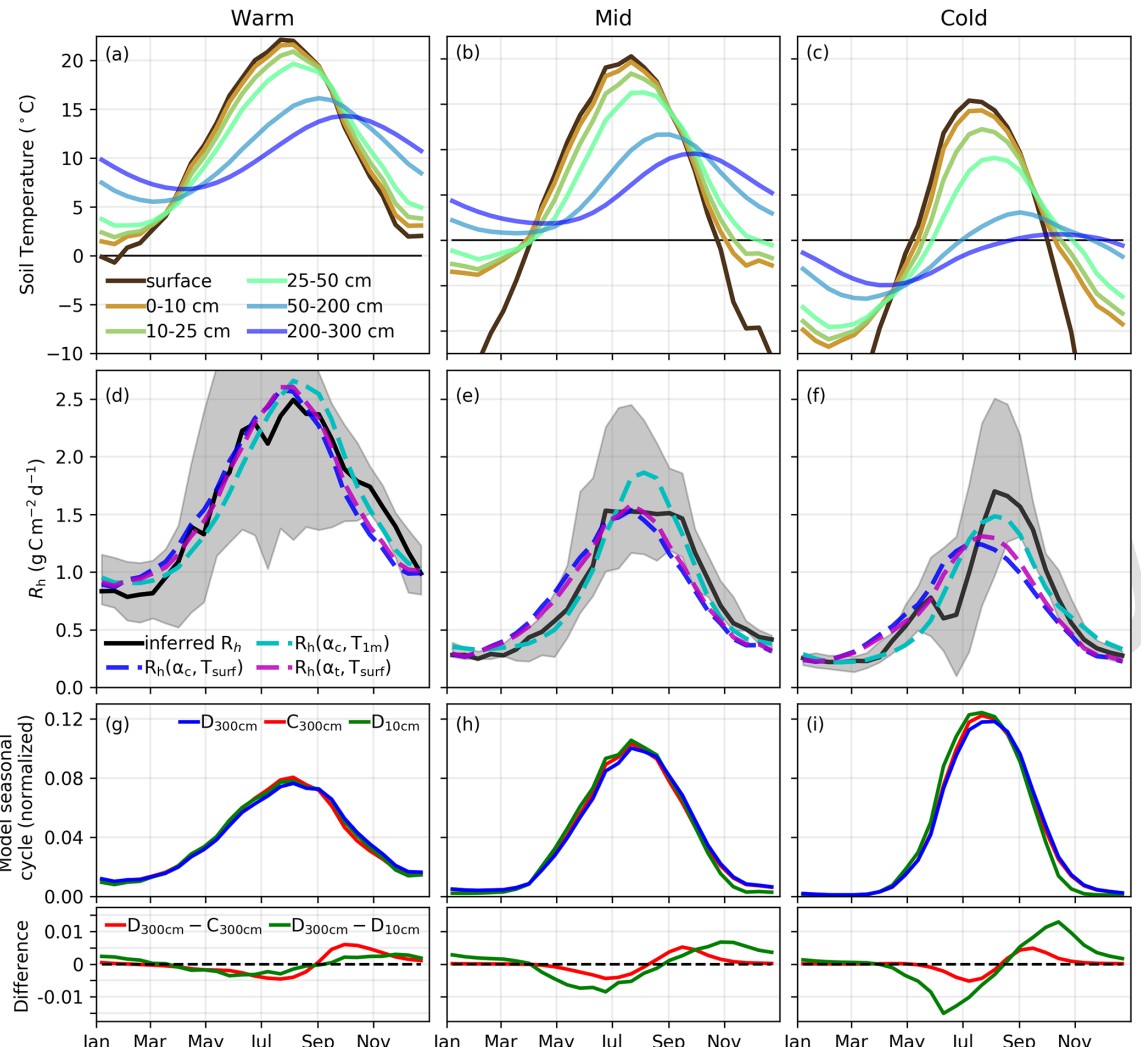

**Figure 5.** TS7 Impact of temporal and vertical variations in carbon pools on $R_h$. **(a–c)** MERRA-2 Land soil temperature over five intervals for the **(a)** Warm, **(b)** Mid, and **(c)** Cold regions. **(d–e)** Mean and range in inferred 14 d $R_h$ with fits for single-layer $R_h$ models that employ (navy dash) $T_{surf}$ dependence and no seasonal variations in the carbon pool, (cyan dash) $T_{1\,m}$ dependence and no seasonal variations in the carbon pool, and (magenta dash) $T_{surf}$ dependence and seasonal variations in the carbon pool. **(g–i**, top) Normalized seasonal cycle of $R_h$ simulated by the soil decomposition model (Sect. 2.6). The different lines show different model simulations: $D_{300\,cm}$ employs a dynamic carbon pool over 0–300 cm depth, $C_{300\,cm}$ employs a constant carbon pool over 0–300 cm depth, and $D_{10\,cm}$ employs a dynamic carbon pool over 0–10 cm depth. **(g–i**, bottom) Differences in simulated $R_h$ between experiments.

We further investigate these mechanisms using a soil carbon decomposition model that can simulate seasonal and vertical variations in carbon pools (Sect. 2.6). This allows for a prognostic simulation of mechanisms driving the seasonality, in contrast to the diagnostic one-layer models. Three experiments are examined that simulate $R_h$: (1) down to a depth of 300 cm using a constant carbon pool ($C_{300\,cm}$), (2) within the top 10 cm of soil using a dynamic carbon pool ($D_{10\,cm}$), and (3) to a depth of 300 cm using a dynamic carbon pool ($D_{300\,cm}$ due to dynamic litterfall inputs and $R_h$ outputs). We compare these seasonal cycles after normalizing by the annual total $R_h$. Figure 5g–i show that incorporating seasonal and vertical variations in the carbon pool results in a phase shift in $R_h$ to later in the year, consistent with the one-layer model results. The simulated impact of these factors is found to be quite small possibly due to underestimation of the impact of seasonal and vertical variations in the carbon pools on $R_h$ in the model. Still, these model simulations can inform the $R_h$ tendencies of these carbon pool variations. Comparing the regions, the impact of seasonal variations in the labile carbon pool are quite similar, with reduced $R_h$ in the summer and increased $R_h$ during the autumn relative to a constant carbon pool. In contrast, the impact of vertically resolved $R_h$ shows differences between the regions, with a small impact for the Warm region but a comparatively large impact for the Cold region. The larger impact over the Cold region is likely

**Table 1.** Parameters ($\alpha$, $\beta$) for single model fits and statistics (slope, intercept, $R^2$, standard error – SE) on the data–model mismatch of these fits TS9.

| Region | Experiment | Slope | Intercept | $R^2$ | Standard error (SE) ($\mathrm{g\,C\,m^{-2}\,d^{-1}}$) |
|--------|-----------|-------|-----------|-------|------------------------|
| Warm | $\alpha e^{\beta T_{\mathrm{surf}}}$ | 0.94 | 0.11 | 0.89 | 0.067 |
| Warm | $\alpha e^{\beta T_{1\,\mathrm{m}}}$ | 0.91 | 0.14 | 0.93 | 0.051 |
| Warm | $\alpha(t) e^{\beta T_{\mathrm{surf}}}$ | 0.94 | 0.10 | 0.92 | 0.057 |
| Mid | $\alpha e^{\beta T_{\mathrm{surf}}}$ | 1.03 | 0.00 | 0.84 | 0.091 |
| Mid | $\alpha e^{\beta T_{1\,\mathrm{m}}}$ | 0.88 | 0.09 | 0.94 | 0.046 |
| Mid | $\alpha(t) e^{\beta T_{\mathrm{surf}}}$ | 1.04 | −0.01 | 0.89 | 0.074 |
| Cold | $\alpha e^{\beta T_{\mathrm{surf}}}$ | 1.08 | −0.01 | 0.66 | 0.16 |
| Cold | $\alpha e^{\beta T_{1\,\mathrm{m}}}$ | 1.04 | −0.01 | 0.88 | 0.08 |
| Cold | $\alpha(t) e^{\beta T_{\mathrm{surf}}}$ | 1.11 | −0.03 | 0.77 | 0.13 |

due to larger carbon pools at depth (Fig. S17), with a possible contribution from regional differences in the thermal gradient with depth (Fig. 5). Similarly, we find that the fractional contribution of subsurface soils to total $R_h$ has larger seasonal variation over the Cold region (Fig. S18). Thus, these results support a substantial contribution of subsurface soil $R_h$ and suggest that an underestimation of this quantity by the DGVMs could explain the data–model differences.

## 4 Discussion

### 4.1 Implications

Over the cold northeastern region of Eurasia, our data-driven $R_h$ seasonal cycle allocates 64 %–70 % of annual $CO_2$ emissions to outside of the summer (August–April) compared to only 52 % of annual $R_h$ emissions allocated by the TRENDY DVGMs to this period. The reason that the TRENDY models do not capture this seasonality is unclear. A plausible explanation is that the TRENDY models do not capture the contribution of subsurface layers to $R_h$, especially during the zero-curtain period. This is clearly the case for the subset of TRENDY models that drive $R_h$ with air temperature. However, it is unclear if this is an important factor for models with more sophisticated soil modules. Surprisingly, a preliminary analysis did not find a relationship between model complexity and agreement with the data-driven estimate. The drivers of differences from the data-driven estimate may differ between models and be impacted by the interplay of litterfall phenology, $R_h$ formulation (Peylin et al., 2005), and number of soil layers, among other factors. Some potential areas of focus for improving models may be gleaned from recent studies. Seiler et al. (2022) suggest that the TRENDY models may systematically underestimate soil organic carbon at high latitudes, which could contribute to an underestimate of subsurface $R_h$ across the models. Endsley et al. (2022) found a similarly phased bias in simulated $R_h$ by the Terrestrial Carbon Flux (TCF) model against flux tower $R_h$ to that reported here. They show that this bias could be largely mitigated by adding seasonally varying litterfall phenology, an $O_2$ diffusion limitation on $R_h$, and a vertically resolved soil decomposition model, suggesting these may be foci for model improvements.

Differences between the data-driven and TRENDY $R_h$ seasonal cycles suggest that DGVMs may be deficient in simulating the response of permafrost-rich ecosystems to climate change, particularly in terms of subsurface $R_h$. Improving DGVM skill in these ecosystems is critical given the rapid northern high-latitude warming and lengthening of the zero-curtain period (Euskirchen et al., 2017; Parazoo et al., 2018b; Chen et al., 2021). The rapid changes in northern Eurasia are illustrated in Fig. 6, which shows the number of months per year that soil temperatures are greater than 0 °C as simulated by a set of CMIP6 models. Soils in the permafrost-rich Cold region are undergoing the most dramatic lengthening of the unfrozen period, particularly at depth (50–200 cm). Under scenario ssp585 (highest emission scenario), these soils are predicted to go from ∼5 months per year with a monthly mean soil temperature above 0 °C during the 20th century to ∼ 11 months per year by 2100. The impact is largest for the Cold region at depth because of the reduced seasonality relative to the surface such that a warming of ∼ 7 °C shifts nearly the entire seasonal cycle above 0 °C at a depth of 50–200 cm (Fig. S19). Such warming would drive the widespread formation of talik, a subsurface layer of perennial thawed soil (Parazoo et al., 2018b), and further enhance $R_h$ at depth.

$R_h$ from sub-surface layers may already be increasing substantially in permafrost regions. Examining the 41-year record of $CO_2$ at Barrow tower, Commane et al. (2017) find that early cold season NEE efflux (October–December) has increased 73.4 % ± 10.8 % over the 1975–2015 period. The standard CAMS IS inversion product similarly suggests an increase in the September–October NEE efflux of ∼ 80 %

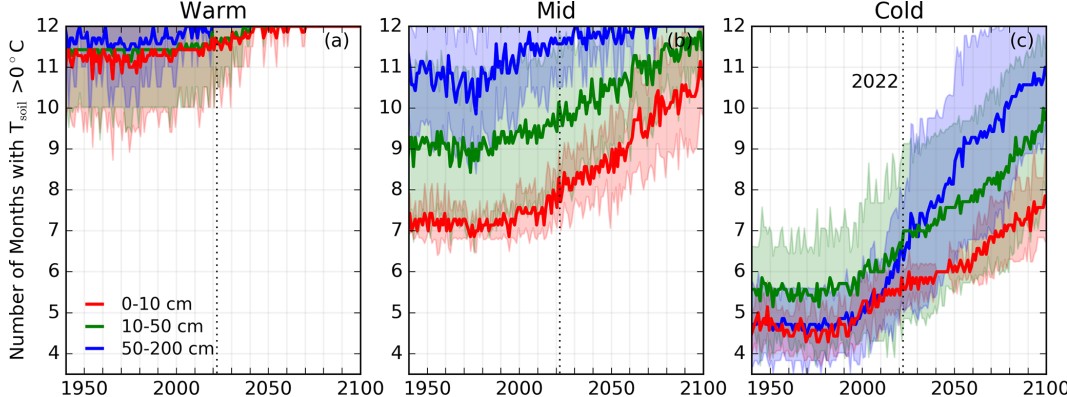

**Figure 6.** Number of months per year with monthly mean soil temperatures above 0 °C at depths of 0–10 cm (red), 10–50 cm (green), and 50–200 cm (blue) simulated by seven CMIP6 models under ssp585 for the **(a)** Warm, **(b)** Mid, and **(c)** Cold regions. The solid lines show the model mean, and shading shows ±1 SD.

over Siberia for the 2013–2017 period relative to the 1980–1984 period (see Fig. S20 of Lin et al., 2020). In agreement, Hu et al. (2021) identified a strong increase ($\sim 10\%$) in August–October $R_h$ over the North America Arctic–boreal region between 1979–1988 and 2010–2019 based on measurements of atmospheric $CO_2$ and carbonyl sulfide. These inferred changes in $R_h$ may in part be related to warming-induced changes in the seasonality of GPP (Liu et al., 2020; Kwon et al., 2021), but more research is needed to determine the impact of these different drivers.

## 4.2 Limitations

Atmospheric $CO_2$ measurements are relatively sparse over northern Eurasia (Byrne et al., 2017). In situ and flask $CO_2$ measurements are spatially sparse over Mid and Cold regions (Fig. S8 TS11), with only a handful of sites assimilated over Russia as part of Japan–Russia Siberian Tall Tower Inland Observation Network (JR-STATION) of nine tower sites (Sasakawa et al., 2010, 2013). The OCO-2 coverage is seasonally variable (Fig. S7 TS12). Due to the fact that $X_{CO_2}$ retrievals are performed on reflected sunlight, the coverage across Eurasia is quite good during the growing season (May–September). However, low signal and the inability to perform retrievals over snow limit the data coverage during the shoulder seasons and winter, resulting in few $X_{CO_2}$ retrievals across the Mid and Cold regions during November–February. Ongoing research to both improve $X_{CO_2}$ quality control filtering at high latitudes (Jacobs et al., 2020; Mendonca et al., 2021) and retrieve $X_{CO_2}$ over snow and ice surfaces (Mikkonen et al., 2021) may reduce these data gaps in the future. Despite this sparsity of measurements, we find that the LNLG and IS flux inversions show consistent differences from the TRENDY and prior fluxes. Furthermore, these data show good agreement with withheld in situ data (Peiro et al., 2022) and independent aircraft measurements over Alaska (Fig. S10 TS13). Thus, we believe the results presented here to be robust despite data gaps. Still, this sparsity of data leads to some limitations. There are few sources of independent $CO_2$ measurements over the Mid and Cold regions to evaluate the inversion posterior $CO_2$ fields. Independent measurements (possibly aircraft campaigns) would provide a valuable additional data set for validation. Similarly, increasing the number of year-round eddy-covariance sites across the Mid and Cold regions would provide a valuable independent dataset to compare against flux inversion estimated NEE. For example, Byrne et al. (2020a) were able to confirm top-down estimates of east–west differences in NEE interannual variability across North America against the dense network of eddy-covariance sites.

We also note that there are challenges in estimating data-driven GPP during the shoulder season due to reduced reflected radiance and snow cover, which impacts the spectral features of the vegetation canopy. Poor quality data, such as snowy and noisy samples, contribute to uncertainty in the timing of shoulder seasons (Wang et al., 2017; Zhang, 2015). In this analysis, we attempted to mitigate this issue through the use of an ensemble of data-driven GPP estimates, but we acknowledge that remaining biases may be present.

Furthermore, the partitioning of NEE into NPP and $R_h$ could be biased if CUE estimates were seasonally biased. We employed TRENDY model CUE to translate data-driven constraints on NEE and GPP into estimates of NPP and $R_h$. Thus, systematic errors across the TRENDY ensemble in CUE could impact conclusions about the relative contributions of errors in NPP and $R_h$. A potential source of bias in CUE could be due to an underestimate of the impact of inhibition of leaf respiration by light (Wehr et al., 2016; Byrne et al., 2018; Keenan et al., 2019; Oikawa et al., 2017). This would result in greater CUE and NPP during June–July relative to the rest of the year, shifting the inferred $R_h$ seasonal cycle earlier, with $R_h$ increased during June–July but decreased elsewhere (Byrne et al., 2018). However, the magnitude of this impact on the ecosystem scale is uncertain, mak-

ing accounting for this phenomenon challenging. Recently, Endsley et al. (2022) found that the inhibition of leaf respiration by light has a relatively modest impact on the seasonality of NPP and $R_h$, suggesting that the results presented here are robust.

There are also remaining challenges in relating the inferred fluxes to underlying processes. Space-based flux constraints do not discriminate between biological and physical processes driving carbon cycle fluxes. It is currently unclear whether the substantial cold season $CO_2$ effluxes across permafrost regions are driven primarily by biological activity or physical processes (Natali et al., 2019; Arndt et al., 2020; Raz-Yaseef et al., 2017). Yet, isolating the primary driver of these fluxes is critical for inferring the sensitivity of $R_h$ to climate change. If the cold season $R_h$ comes from the metabolism of old permafrost carbon, then $^{14}CO_2$ measurements could help differentiate biological from physical $CO_2$ production.

## 5 Conclusions

Space-based and in situ atmospheric $CO_2$ measurements revealed strong summer uptake and early cold season release of $CO_2$ over the cold northeastern Eurasia region, implying a late summer peak in $R_h$ with substantial early cold season respiration. Based on model simulations of $R_h$, we suggested that this seasonality is driven by a large contribution of subsurface soils to the total $R_h$. These results are consistent with site-level observations identifying substantial $CO_2$ release in permafrost regions outside the growing season (Natali et al., 2019) and, in particular, reported spikes in early cold season respiration associated with the zero-curtain period in Arctic ecosystems (Commane et al., 2017; Jeong et al., 2018).

The data-driven seasonality of $R_h$ over the Cold region was generally not captured by the TRENDY DGVMs, which showed greater $R_h$ during May–July and lower $R_h$ during the rest of the year. The underlying cause of this discrepancy is unclear but may be linked to an underestimate of the contribution of sub-surface soils to total $R_h$. Given the rapid warming of permafrost soils (Euskirchen et al., 2017; Chen et al., 2021), talik formation (Parazoo et al., 2018b), and increasing early cold season $CO_2$ effluxes (Commane et al., 2017; Lin et al., 2020; Hu et al., 2021), improving DGVM simulations in permafrost regions should be a focus of future studies.

This analysis demonstrates the utility of space-based observations for studying carbon cycle dynamics at high latitudes, where in situ measurements are sparse. Although currently limited by a short observing record (2014–present), the estimates of NEE inferred from the OCO-2 $X_{CO_2}$ retrievals suggest that these data will provide a powerful tool for detecting change in seasonal cycle of NEE across northern Eurasia.

## Appendix A: Appendix 1

We estimate seasonal variations in labile carbon by estimating a litterfall flux of carbon. Litterfall seasonality is assumed to follow the same pattern as Randerson et al. (1996) (Fig. S14 TS14). We assume that the labile carbon pool is in steady state on annual timescales such that the annual total literfall is equal to the annual total $R_h$:

$$\text{Litterfall}(t) = f_{\text{NPP}}(t) \int_0^{365} R_h(t) \, dt, \tag{A1}$$

where $t$ is the day of the year, and $f_{\text{NPP}}$ is the fraction of annual total NPP that is converted to litterfall. The seasonal variation in the labile carbon pool ($\Delta C_{\text{pool}}$) is defined as the difference in flux between litterfall and $R_h$:

$$\Delta C_{\text{pool}}(t) = \int_0^t (\text{Litterfall}(t) - R_h(t)) \, dt. \tag{A2}$$

Finally, we assume a fractional variation in the total carbon pool amount, $\gamma$, and calculate $\alpha(t)$:

$$\alpha(t) = \left( \frac{C_{\text{pool}}(t)}{\max\left(|C_{\text{pool}}(t)|\right)} \gamma + 1 \right) \alpha_0, \tag{A3}$$

where $\alpha_0$ is the mean carbon pool size and is optimized in the regression in Sect. 3.3.

*Data availability.* TS15 TRENDY v8 gridded data were accessed by contacting Stephen Sitch following the TRENDY data policy described on their website: https://sites.exeter.ac.uk/trendy (Sitch et al., 2022). v9 OCO-2 MIP fluxes were downloaded from https://gml.noaa.gov/ccgg/OCO2_v9mip/ (Crowell et al., 2022). GFED data were downloaded from https://www.globalfiredata.org/ (Randerson et al., 2022). We downloaded version 10 of the ACOS OCO-2 lite files from the GES DISC (https://doi.org/10.5067/W8QGIYNKS3JC, OCO-2 et al., 2018). OCO-2 data were produced by the OCO-2 project at the Jet Propulsion Laboratory, California Institute of Technology, and obtained from the OCO-2 data archive maintained at the NASA Goddard Earth Science Data and Information Services Center. FluxSat data were downloaded from https://avdc.gsfc.nasa.gov/pub/tmp/FluxSat_GPP/ (Joiner, 2022). The GOSIF data product is available at http://data.globalecology.unh.edu/, (Xiao and Xing, 2022). ERA5-Land data are obtained from the Climate Data Store (https://doi.org/10.24381/cds.68d2bb30, Muñoz Sabater, 2019).

*Supplement.* The supplement related to this article is available online at: https://doi.org/10.5194/bg-19-1-2022-supplement. TS16

*Author contributions.* BB, JL, YY, AC, KWB, NCP, DC, and CEM conceived of the study. BB, JL, YY, AC, and SB designed the experiments. YY performed the soil carbon decomposition model runs.

BB, SB, and FC performed inversions for this study. BB, RC, and RD performed analysis of GPP data. XL and JX created GOSIF data. SS, BG, and PCM performed TRENDY simulations. BB created the figures and prepared the manuscript with contributions from all co-authors.

*Competing interests.* The contact author has declared that none of the authors has any competing interests.

*Acknowledgements.* Brendan Byrne and Junjie Liu were supported by the NASA OCO2/3 science team program NNH17ZDA001N-OCO2. Abhishek Chatterjee, Brendan Byrne, Junjie Liu, and Sourish Basu were also supported by the NASA OCO Science Team Grant #80NSSC21K1068. Charles E. Miller was supported by NASA's Arctic Boreal Vulnerability Experiment (ABoVE) under NNH18ZDA001N-TE. Jingfeng Xiao was supported by the National Science Foundation (NSF) (Macrosystem Biology & NEON-Enabled Science program: DEB-2017870). Matthew S. Johnson acknowledges the internal funding from NASA's Earth Science Research and Analysis Program. Sajeev Philip acknowledges financial support of the NASA Academic Mission Services by Universities Space Research Association at NASA Ames Research Center. Frédéric Chevallier was funded by the Copernicus Atmosphere Monitoring Service, implemented by the European Centre for Medium-Range Weather Forecasts on behalf of the European Commission (grant no. CAMS73). The research carried out at the Jet Propulsion Laboratory, California Institute of Technology, was under a contract with the National Aeronautics and Space Administration. Resources supporting this work were provided by the NASA High-End Computing (HEC) program through the NASA Advanced Supercomputing (NAS) Division at Ames Research Center. Frédéric Chevallier was granted access to the HPC resources of TGCC under the allocation A0110102201 made by GENCI. The ODIAC project is supported by Greenhouse Gas Observing SATellite (GOSAT) project, National Institute for Environmental Studies (NIES), Japan.

*Financial support.* This research has been supported by the National Aeronautics and Space Administration (grant nos. 80NSSC21K1068, NNH17ZDA001N-OCO2, and NNH18ZDA001N-TE).

*Review statement.* This paper was edited by David Bowling and reviewed by Ashley Ballantyne and one anonymous referee.

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

**Remarks from the typesetter**

TS1    Please confirm.

TS2    Please confirm.

TS3    Please confirm.

TS4    Please confirm.

TS5    Please confirm newly inserted figure.

TS6    Please confirm.

TS7    Please confirm newly inserted figure.

TS8    Please confirm.

TS9    Thank you for the explanation given. I just wanted to ask the handling editor for approval of the requested changes but it is very confusing as in the last Proofreading PDF you asked us to change the caption (we did) but the newly provided file for Table 1 'bg-2022-40-Table1_ complete. png" has a different caption. Please let me know if this file is correct with the caption as it is now or does it need a different caption?

TS10    Please confirm.

TS11    Please confirm.

TS12    Please confirm.

TS13    Please confirm.

TS14    Please confirm.

TS15    Please check the data availability section throughout carefully and confirm.

TS16    Thank you. The new Supplement was uploaded.

TS17    Please confirm.

TS18    Please confirm.

TS19    Please confirm.