# Peer review of "Multi-year observations reveal a larger than expected autumn respiration signal across northeast Eurasia"

_Biogeosciences, 2022_

## Referee Comment (RC1)

[referee-annotated manuscript omitted]

---

## Author Comment (AC1)

Dear Dr. Bowling,

We appreciated the constructive comments of the reviewers. We have addressed the comments below. Reviewer/editor comments are shown in bold with our responses in blue. Line numbers refer to the tracked changes manuscript, and changes to the text are underlined.

**Reviewer 1**

**Summary:**

**In this paper Byrne et al. evaluate the seasonal distribution of NEE at high latitudes using a combination of atmospheric CO2 measurements to inform model inversions of net C exchange in combination with satellite estimates of GPP to infer respiration. They note that anomalously low NEE in autumn can be attributed to greater Rh release. They also note a mismatch between their estimates and those derived from land surface models. They then provide an explanation whereby temperature lags within the soil can explain a certain fraction of this enhanced autumn respiration. This was a nice paper and could be publishable with some additional analysis and consideration of assumptions.**

**General Comments:**

**This paper profits from the high resolution XCO2 measurements which now allow us to estimate net CO2 fluxes at high resolution within specific bioregions and uncover different processes that may be affecting these seasonal fluxes. I found this analysis to be quite thorough and pretty convincing; however, I did have a few general comments. It seems as though the mismatch between observations and models may be dependent on the seasonal estimates of CUE and their assumptions. Perhaps it would be useful to look at total ecosystem respiration (GPP-NEE= TER) initially to see if the same mismatch is evident, this would suggest that the mismatch is not an artifact of the unique CUE applied over this region. Alternatively, one could use an independent estimate of CUE from an independent model (Konings et al. 2019) or use the same seasonal CUE for all regions.**

We agree that the CUE is a very important parameter in decomposing NEE between NPP and Rh, and that there is considerable uncertainty in this value. We attempted to minimize the impact of this by applying the CUE estimates from the TRENDY ensemble, and thus applying the same CUE estimate to go from GPP to NEE (on average). We also propagated uncertainty from CUE (based on the TRENDY ensemble spread) into the NPP and Rh estimates (described in Sec. 2.4 and Sec. 2.5). Shortcomings of this approach are also discussed in Sec. 4.2 (L445-454). Still, we agree that it would be useful to the reader to present the more direct GPP and TER estimates. Therefore, we have added a plot of these fluxes to Sec. 3.1 (revised Fig. 2) along with the following accompanying text:

L257-271: "To further investigate the causes of differences in NEE between the TRENDY and v9 OCO-2 MIP ensembles, we separately examine component primary productivity and respiration fluxes. As the most direct decomposition, we employ the data-driven GPP estimates to decompose NEE into GPP and terrestrial ecosystem respiration fluxes (TER) (Fig. 2). This

comparison shows that the TRENDY ensemble mean GPP tends to overestimate the data-driven GPP during the autumn (Sep–Nov), largely explaining the mismatch in NEE during this season. For TER, we find good agreement for over the Warm region except for an underestimate of TER for the TRENDY ensemble mean during the summer (mirroring GPP). For the Mid regions, agreement is found between the TRENDY and data-driven TER estimates throughout the growing season. For the Cold region, we find that the TRENDY ensemble mean suggested greater TER during May–Aug, which drives the mismatch found in NEE.

We next decompose NEE into component NPP and Rh fluxes. These estimates require an additional assumption about the CUE in comparison to the GPP/TER decomposition, but also have the potential to provide more process understanding. As described in Sec. 2.4, we employ the monthly CUE estimates from the ensemble of TRENDY models, this both allows an "apples-to-apples" comparison with the TRENDY models as the CUE estimates are consistent between the data-driven and TRENDY estimates, and allows us to propagate uncertainty in CUE from the ensemble spread. The data-driven NPP and TRENDY NPP are shown in Fig. 3(d-f). The seasonality in NPP between the data-driven and TRENDY estimates show good agreement for all regions.

[Figure]

Figure 2. Monthly carbon cycle fluxes (average of 2015, 2016 and 2018; 2017 is excluded due to an OCO-2 data gap). (a-c) Mean (solid line) and interquartile range (shaded area) of NEE for the ensemble of IS (red) and LNLG (blue) v9 OCO-2 MIP and for the TRENDY ensemble (green). (d-f) GPP for the TRENDY ensemble (green) and data-driven datasets (black). (g-i) TER simulated by the TRENDY ensemble (green) and calculated from combining the data-driven GPP with the IS (red) and LNLG (blue) v9 OCO-2 MIP NEE constraints."

**Furthermore, see recent analysis on Siberian warming where there is a strong relationship between spring GPP and fall TER (Kwon et al. 2021). Although the timespan for the OCO-2 inversions is too short, this citation on seasonal anomalies may help to put these results in a longer temporal context.**

We have noted this and added a reference in the Discussion section (Sec. 4.1):

L422-424: "These inferred changes in Rh may in part be related to warming-induced changes in the seasonality of GPP (Liu et al., 2020; Kwon et al., 2021), but more research is needed to determine the impact of these different drivers."

**I also had some comments on the soil model comparisons with the observation constrained estimates. The text (line 323) discusses regressions and statistics of those regressions, but the actual figure shows seasonal distributions from models and observations. The figure is pretty clean and easy to interpret, but the paper could benefit from a table in the main text that include your statistics, in addition to standard model performance statistics such as RMSE, MAE, and bias statistics.**

We have added Table 1 to the manuscript that provides statistics on the single-layer model fits.

**Table 1.** Statistics on the data-model fits for the single layer models.

| Region | Experiment | Slope | Intercept $(gCm^{-2}day^{-1})$ | $R^2$ | Standard Error (SE) $(gCm^{-2}day^{-1})$ |
|--------|-----------|-------|-----------|-------|----------------------|
| Warm | $\alpha e^{\beta T_{surf}}$ | 0.94 | 0.11 | 0.89 | 0.067 |
| Warm | $\alpha e^{\beta T_{1m}}$ | 0.91 | 0.14 | 0.93 | 0.051 |
| Warm | $\alpha(t)e^{\beta T_{surf}}$ | 0.94 | 0.10 | 0.92 | 0.057 |
| Mid | $\alpha e^{\beta T_{surf}}$ | 1.03 | 0.00 | 0.84 | 0.091 |
| Mid | $\alpha e^{\beta T_{1m}}$ | 0.88 | 0.09 | 0.94 | 0.046 |
| Mid | $\alpha(t)e^{\beta T_{surf}}$ | 1.04 | -0.01 | 0.89 | 0.074 |
| Cold | $\alpha e^{\beta T_{surf}}$ | 1.08 | -0.01 | 0.66 | 0.16 |
| Cold | $\alpha e^{\beta T_{1m}}$ | 1.04 | -0.01 | 0.88 | 0.08 |
| Cold | $\alpha(t)e^{\beta T_{surf}}$ | 1.11 | -0.03 | 0.77 | 0.13 |

**The model could also be tested against the eddy flux data estimates of Rh and these values could be reported in the table. This would help the reader evaluate which models are indeed superior.**

We have added fits to the FLUXNET seasonal cycle of Rh. We find consistent results with the analysis for the cold region. We have added a supplementary figure and table to show these results, and have added the following text to the main manuscript:

L360-365: "To further confirm that $R_h(\alpha_c, T_{1m})$ best captures the seasonality of $R_h$, we fit these same models to seasonal FLUXNET $R_h$ averaged over cold sites. This is a rather rough comparison as we drive the models with soil temperatures averaged over the Cold region rather than site specific datasets (due to absence of soil temperature data). Figure S16 shows the resulting fits and Table S2 gives the statistics of the fits. We find that $R_h(\alpha_c, T_{1m})$ performs best ($R^2 = 0.96$, SE $= 0.08$ gCm$^{-2}$ day$^{-1}$), while $R_h(\alpha_t, T_{surf})$ performs second best ($R^2 = 0.85$, SE $= 0.19$ gCm$^{-2}$ day$^{-1}$) and $R_h(\alpha_c, T_{surf})$ gives the poorest performance ($R^2 = 0.75$, SE $= 0.25$ gCm$^{-2}$ day$^{-1}$), consistent with the regional-scale data-driven analysis."

**Table S2.** Statistics on the data-model fits for the single layer models against the FLUXNET inferred $R_h$ seasonal cycle.

| Experiment | Slope | Intercept (gCm$^{-2}$day$^{-1}$) | $R^2$ | Standard Error (gCm$^{-2}$day$^{-1}$) |
|---|---|---|---|---|
| $\alpha e^{\beta T_{surf}}$ | 1.36 | -0.14 | 0.75 | 0.25 |
| $\alpha e^{\beta T_{1m}}$ | 1.28 | -0.14 | 0.96 | 0.08 |
| $\alpha(t)e^{\beta T_{surf}}$ | 1.4 | -0.17 | 0.85 | 0.19 |

[Figure]

**Figure S13.** Mean and range in inferred monthly FLUXNET $R_h$ with fits for single-layer $R_h$ models that employ (navy dash) $T_{surf}$ dependence and no seasonal variations in the carbon pool, (cyan dash) $T_{1m}$ dependence and no seasonal variations in the carbon pool and (magenta dash) $T_{surf}$ dependence and seasonal variations in the carbon pool.

**Also litterfall estimates seem an order of magnitude too high in Fig. s15 should peak at ~2 TgC day-1 as compared to NPP estimates in Fig. 3. This may just be a units problem but check the model.**

We had an error in the model and description. The annual total litterfall is defined to be equal to the annual total Rh, so that the labile carbon pool is in steady state. Thus, the monthly litterfacll

fluxes are equal to "f$_{npp}$*sum(R$_h$)", where s sum(R$_h$) is the total Rh over a year. Please see the revised Appendix 1. The corrected figure is shown below (note that changes are very small).

[Figure]

**Figure S16.** (top) Fraction of NPP that becomes litterfall. (middle row) Carbon flux from litterall. (bottom) Seasonal variations in the labile carbon pool due to litterfall and R$_h$.

**Specific comments:**

**L26: more of a synthesis than a conventional meta-analysis**

Fixed

**L46: sparse**

Fixed

**L47: Comane et al. used a top-down approach as well, but no satellite estimates based on my recollection**

Yes, that is correct. They perform a more regional analysis with measurements from Barrow tower and aircraft campaigns.

**L51: seems like you need more of a justification of why Eurasia other than just the sparsity of data. See Bastos et al. on attribution of the enhanced seasonal cycle to Eurasia: https://doi.org/10.5194/acp-19-12361-2019**

Added additional context:

L50-53: "We utilize these data to investigate carbon cycle dynamics over three large regions within Eurasia (Fig. 1), which are defined based on the east-west temperature gradient (see Sec. 2.1), with the coldest region in the east and warmest region in the west. We focus on Eurasia as much of this region has particularly sparse site-level observations yet is experiencing rapid change (Liu et al., 2020; Bastos et al., 2019)."

**L90: what is this resolution state here.**

The spatial resolution differs for each dataset, that is why we state them when introducing the specific datasets.

**L145: are all these soundings weighted equally or do some higher error or bias?**

Yes, clarified.

**L156: why did you include land use? This seems unnecesary for your research questions and adds a potentially confounding factor.**

The data-driven estimates of NEE and GPP will be impacted by land use, therefore we compare to this experiment as it relates most closely to reality.

**L163-164: how much spatial variability is there among these CUE estimates?**

This can be seen from comparing the regions in Fig. S5. The CUE falls to a similar value of ~0.5 during the growing season.

**Eq 4: omit already given as equation 1**

Done

**L206-207: this of course could change considerably with surface water pooling happens above poorly drained permafrost soils**

Yes, changes in the thermal regime could have these knock-on impacts.

**L218-219: once again depend on the CUE estimates and assumptions**

Please see the response to the general comments

**L253: NPP estimates largely**

fixed

**L257-258: of course all of the unexplained variance is ascribed to the inferred term in the budget or Rh**

This is correct. However, we propagate uncertainties in each quantity (GPP, NEE, CUE) such that systematic differences should be attributable to Rh.

**L297-299: this dip in respiration could be due to the timing of snow melt as well that insulates the soil and promotes Rh: https://doi.org/10.1111/geb.12441**

We have added this as a possible mechanism to the manuscript:

L321-324: "A potential mechanistic explanation for a spring pulse of Rh could be due to thawing soils that release $CO_2$ that has been trapped within subsurface soil layers over the winter (see Sec. 4). Another plausible mechanism could be the timing of snow melt, which may insulate the soil over winter (Yu et al., 2016). However, the signal from this first peak is small relative to the uncertainties."

**L318: should probably only consider the top 0.5 m considering below that is below zero year round in the cold region**

We feel that using the top 1m is a reasonable average. The temperature is not below zero year-round for 50-100 cm. In fact, even the 50-200 cm interval is above zero from Jul-Nov, as shown in Fig. 4.

**L322: these just seem to be the seasonal cycles with no regression or RMSE statistics reported**

We have added regression statistics as described in response to the general comments.

**L326: it would be helpful if you used the same terms to describe the same experiments in the figure and the text.**

We have revised this section to give consistent model names in the figure and text. The experiments are now named "$R_h(\alpha_c, T_{surf})$", "$R_h(\alpha_c, T_{1m})$", and "$R_h(\alpha_t, T_{surf})$".

**L339-340: do they incorporate differences in litterfall as a function of NPP?**

Yes, we have clarified this in the text:

L375: "… and simulated to a depth of 300 cm using a dynamic carbon pool (D300cm, due to dynamic litterfall inputs and Rh outputs)."

**L357-359: this could be tested with more years of data where litterfall may vary as a function of inter-annual variability in NPP see Kwon et al. https://doi.org/10.1088/1748-9326/ac358b**

Agreed, however, we have a relatively temporally limited record from OCO-2. Further, we find that the sensitivity of these inversions to interannual variations in NEE at high latitudes was quite limited (not shown).

**L422-423: this is a good idea. Have Schur et al. measured 14CO2 seasonally or just annually?**

We are unaware of the density of existing $^{14}$C measurements at high latitudes, but we agree that this would be something to follow-up on in future research.

**Overall, I think the comparison between inversion results and models is really useful, and the paper should be published. But I find it an interesting but not entirely satisfying analysis. One problem is that the number of different steps from NEE to Rh seems like it introduces the potential for several errors to creep in, particularly as relate to Ra.**

We agree that the CUE is a very important parameter in decomposing NEE between NPP and Rh, and that there is considerable uncertainty in this value. We attempted to minimize the impact of this by applying the CUE estimates from the TRENDY ensemble, and thus applying the same CUE estimate to go from GPP to NEE (on average). We also propagated uncertainty from CUE (based on the TRENDY ensemble spread) into the NPP and Rh estimates (described in Sec. 2.4 and Sec. 2.5). Shortcomings of this approach are also discussed in Sec. 4.2 (L445-454). Still, we agree that it would be useful to the reader to present the more direct GPP and TER estimates. Therefore, we have added a plot of these fluxes to Sec. 3.1 (revised Fig. 2) along with the following accompanying text:

L257-271: "To further investigate the causes of differences in NEE between the TRENDY and v9 OCO-2 MIP ensembles, we separately examine component primary productivity and respiration fluxes. As the most direct decomposition, we employ the data-driven GPP estimates to decompose NEE into GPP and terrestrial ecosystem respiration fluxes (TER) (Fig. 2). This comparison shows that the TRENDY ensemble mean GPP tends to overestimate the data-driven GPP during the autumn (Sep–Nov), largely explaining the mismatch in NEE during this season. For TER, we find good agreement for over the Warm region except for an underestimate of TER for the TRENDY ensemble mean during the summer (mirroring GPP). For the Mid regions, agreement is found between the TRENDY and data-driven TER estimates throughout the growing season. For the Cold region, we find that the TRENDY ensemble mean suggested greater TER during May–Aug, which drives the mismatch found in NEE.

We next decompose NEE into component NPP and Rh fluxes. These estimates require an additional assumption about the CUE in comparison to the GPP/TER decomposition, but also have the potential to provide more process understanding. As described in Sec. 2.4, we employ the monthly CUE estimates from the ensemble of TRENDY models, this both allows an "apples-to-apples" comparison with the TRENDY models as the CUE estimates are consistent between the data-driven and TRENDY estimates, and allows us to propagate uncertainty in CUE from the ensemble spread. The data-driven NPP and TRENDY NPP are shown in Fig. 3(d-f). The seasonality in NPP between the data-driven and TRENDY estimates show good agreement for all regions.

[Figure]

Figure 2. Monthly carbon cycle fluxes (average of 2015, 2016 and 2018; 2017 is excluded due to an OCO-2 data gap). (a-c) Mean (solid line) and interquartile range (shaded area) of NEE for the ensemble of IS (red) and LNLG (blue) v9 OCO-2 MIP and for the TRENDY ensemble (green). (d-f) GPP for the TRENDY ensemble (green) and data-driven datasets (black). (g-i) TER simulated by the TRENDY ensemble (green) and calculated from combining the data-driven GPP with the IS (red) and LNLG (blue) v9 OCO-2 MIP NEE constraints."

**Second, there are any number of reasons why the DGVMs could show a bias relative to the observations, and it is certainly possible that the lack of deep-soil respiration is one reason. But the attempt to provide a mechanistic explanation here using a simple model is not very clear, and subject to somewhat arbitrary choices like how to handle substrate seasonality. I wonder if a slightly different approach of looking at the DGVMs themselves, and asking whether there are structural or parametric characteristics of the models that govern the shapes of their seasonal cycles, and which might provide some clues for identifying whether any of them do a better or worse job than others?**

We agree that it would be preferable to identify model differences across the TRENDY ensemble that can explain the data-model differences. We attempted to perform such an analysis but it was unsuccessful. One of the challenges is that the models general differ in many ways, thus isolating

specific factors in very challenging. This is why we have adopted the simpler approach of using an idealized one-layer diagnostic model, and one diagnostic model run with different set-ups. We address the challenges if identifying mechanisms driving these differences across the TRENDY ensemble in the discussion section:

L390-404: "Over the cold northeastern region of Eurasia, our data-driven Rh seasonal cycle allocates 64–70% of annual $CO_2$ emissions to outside of the summer (August - April) compared to only 52% of annual Rh emissions allocated by the TRENDY DVGMs to this period. The reason that the TRENDY models do not capture this seasonality is unclear. A plausible explanation is that the TRENDY models do not capture the contribution of subsurface layers to $R_h$, especially during the zero-curtain period. This is clearly the case for the subset of TRENDY models that drive $R_h$ with air temperature. However, it is unclear if this is an important factor for models with more sophisticated soil modules. Surprisingly, a preliminary analysis did not find a relationship between model complexity and agreement with the data-driven estimate. The drivers of differences from the data-driven estimate may differ between models, and be impacted by the interplay of litterfall phenology, $R_h$ formulation (Peylin et al., 2005), and number of soil layers, among other factors. Some potential areas of focus for improving models may be gleaned from recent studies. Seiler et al. (2021) suggest that the TRENDY models may systematically underestimate soil organic carbon at high latitudes, which could contribute to an underestimate of subsurface $R_h$ across the models. Endsley et al. (2021) found a similarly phased bias in simulated $R_h$ by the Terrestrial Carbon Flux (TCF) model against flux tower $R_h$ to that reported here. They show that this bias could be largely mitigated by adding seasonally varying litterfall phenology, an $O_2$ diffusion limitation on $R_h$ and a vertically resolved soil decomposition model, suggesting these may be foci for model improvements."

**Line 12: please provide uncertainty range for the DGVM estimates, as you do for the data-driven estimates**

Done:

Furthermore, we show that this seasonality of NEE and Rh over northeastern Eurasia is not captured by the TRENDY v8 ensemble of dynamic global vegetation models (DGVMs), which estimate that 47–57% (interquartile range) of annual Rh occurs during Aug-Apr, while the data-driven estimates suggest 59–76% of annual Rh occurs over this period.

**Line 17: "is not well captured by current DGVMs." Any DGVMs, or just the ensemble mean?**

We believe the current text, "… suggests that autumn Rh from subsurface soils in the northern high latitudes is not well captured by current DGVMs", to be accurate. The differences examined here are generally considered within uncertainties based on the interquartile range, thus we could consider differences to be robust at this level. We did look at individual models when trying to better understand mechanisms (not reported in the study due to inconclusive results), and all models showed deficiencies in reproducing the data-driven Rh seasonal cycle across the regions, consistent with this general statement.

**Line 70-72: Could you clarify whether you are using monthly mean CUE values or annual mean values here?**

Monthly, clarified.

**Lines 99-100. How confident are we in the soil temperature predictions of these models? There have been a few analyses of the soil temperature dynamics and permafrost statistics of climate models at high latitudes. Does this set represent a set of best-performing soil temperature models?**

We compared the ensemble to the reanalysis datasets from ERA5 and MERRA2, and they generally reproduced similar seasonality (Fig. S14). We also represent the interquartile spread among the models, shown by the shaded area.

In addition, we compared the MERRA2 soil temperature seasonality to borehole measurements (Text S2 and Fig. S13) and found good agreement overall. Given that the models generally show agreement with MERRA2 soil temperature, we can conclude that the model seasonality is largely consistent with the borehole data as-well.

**Fig. 2. I think that the per-area fluxes are more meaningful here, otherwise the reader gets the suggestion that NEE is higher during the summer in the colder than the warmer regions, which is confusing. So I suggest switching fig. S7 and fig. 2, and in general reporting things per unit area.**

We have changed the reported units to gC m$^{-2}$ day$^{-1}$.

**Figs 2 & S7: I am skeptical about the errors introduced by the GPP -> NPP conversion, I think it would be useful to include a set of GPP panels as well, since, like NEE, that is the most directly observed, with the NPP and RH much less direct.**

Please see our response to the first comment.

**Figs 2 and S7: A lot of the focus of the discussion is on the autumn differences, but I wonder if the more general problem is that the winter respiration in general is underestimated by the models in the cold region. This would be consistent with the findings of Natali et al., but given the larger-scale datasets used here would still be an important point to emphasize here.**

We have decided against emphasizing differences during the winter as this season is less well informed by the CO$_2$ measurements due to small overall fluxes and sparse sampling (e.g., Byrne et al., 2017). Thus, some of the differences in the winter might be impacted by differences in the prior fluxes rather than being informed from the observational constraints. Furthermore, the absolute differences in NEE are much larger during the growing season.

Byrne, B., Jones, D. B. A., Strong, K., Zeng, Z.-C., Deng, F., and Liu, J.: Sensitivity of $CO_2$ Surface Flux Constraints to Observational Coverage, J. Geophys. Res.-Atmos, 112, 6672–6694, https://doi.org/10.1002/2016JD026164, 2017

**Lne 264: This isn't really a shift, so much as a bias in TRENDY relative to the observations?**

We were referring to a shift in the seasonality, that is, a large fraction of Rh occurs during May-Jul relative to the rest of the year.

**Lines 264-270 and fig. S12. I think FLUXNET is actually telling a different story than the larger-scale datasets. The TRENDY models actually have a higher positive NEE anomaly during the shoulder season than FLUXNET, which is the opposite pattern shown in fig. 2c.  If this is correct, then I think the discussion of this result needs to be revised accordingly.**

It is true that there are some differences in the NEE and GPP relative to the regional estimates. However, the we are referring specifically to the $R_h$ seasonality, which peaks later in the year than the TRENDY ensemble predicts. In the revised manuscript, we have added fits with the one-layer model to the FLUXNET-based $R_h$ seasonal cycle, and find consistent results with the analysis for the cold region. We have added a supplementary figure and table to show these results, and have added the following text to the main manuscript:

L360-365: "To further confirm that $R_h(\alpha_c, T_{1m})$ best captures the seasonality of $R_h$, we fit these same models to seasonal FLUXNET $R_h$ averaged over cold sites. This is a rather rough comparison as we drive the models with soil temperatures averaged over the Cold region rather than site specific datasets (due to absence of soil temperature data). Figure S16 shows the resulting fits and Table S2 gives the statistics of the fits. We find that $R_h(\alpha_c, T_{1m})$ performs best ($R^2 = 0.96$, SE = 0.08 $gCm^{-2}$ $day^{-1}$), while $R_h(\alpha_t, T_{surf})$ performs second best ($R^2 = 0.85$, SE = 0.19 $gCm^{-2}$ $day^{-1}$) and $R_h(\alpha_c, T_{surf})$ gives the poorest performance ($R^2 = 0.75$, SE = 0.25 $gCm^{-2}$ $day^{-1}$), consistent with the regional-scale data-driven analysis."

**Table S2.** Statistics on the data-model fits for the single layer models against the FLUXNET inferred $R_h$ seasonal cycle.

| Experiment | Slope | Intercept ($gCm^{-2}day^{-1}$) | $R^2$ | Standard Error ($gCm^{-2}day^{-1}$) |
|---|---|---|---|---|
| $\alpha e^{\beta T_{surf}}$ | 1.36 | -0.14 | 0.75 | 0.25 |
| $\alpha e^{\beta T_{1m}}$ | 1.28 | -0.14 | 0.96 | 0.08 |
| $\alpha(t) e^{\beta T_{surf}}$ | 1.4 | -0.17 | 0.85 | 0.19 |

[Figure]

**Figure S13.** Mean and range in inferred monthly FLUXNET $R_h$ with fits for single-layer $R_h$ models that employ (navy dash) $T_{surf}$ dependence and no seasonal variations in the carbon pool, (cyan dash) $T_{1m}$ dependence and no seasonal variations in the carbon pool and (magenta dash) $T_{surf}$ dependence and seasonal variations in the carbon pool.

**Section 3.2, I'm not sure I understand what new information the 14-day-resolved data provides beyond what is in the monthly data. Is this analysis really necessary? If so, could the authors give a bit better motivation and explanation?**

We describe the motivation for this as:

L303-304: "This higher resolution better resolves temporal changes in $CO_2$ fluxes throughout the growing season, particularly during the shoulder seasons, when week-to-week changes in $CO_2$ fluxes are large (Parazoo et al., 2018a)."

We believe that this higher resolution is worthwhile to better resolve variations in the seasonal cycle of NEE, NPP and Rh.

**Fig.3. I'm very skeptical about how narrow the range of uncertainty in panels a-c are here. What is that a measure of?**

The shaded regions are showing the range between three different flux inversions, and is stated in the Figure 3 caption "Median and ensemble spread…". It is a measure of the precision with which flux inversion analyses can estimate fluxes over these regions.

**Lines 334-348, and figure 4. I don't understand this sensitivity analysis, or why the seasonal cycles in panels g-i are so different from the ones in panels d-f. Could you clarify a bit more what is being shown here?**

We have added additional text to better motivate this experiment:

L371-373: "We further investigate these mechanisms using a soil carbon decomposition model that can simulate seasonal and vertical variations in carbon pools (Sec. 2.6). This allows for a prognostic simulation of mechanisms driving the seasonality, in contrast to the diagnostic one-layer models."

And have refined the caption to better explain the experiments:

"… (g-i, top) Normalized seasonal cycle of Rh simulated by the soil decomposition model (Sec. 2.6). The different lines show different model simulations: D300cm employs a dynamic carbon pool over 0-300 cm depth, C300cm employs a constant carbon pool over 0-300 cm depth, D10cm employs a dynamic carbon pool over 0-10 cm depth). (g-i, bottom) Differences in simulated Rh between experiments."

**Further, the argument about deep soil playing a greater role should help with the autumn respiration peak, but less so with the bias in respiration in the cold region throughout the winter. What does this analysis have to say about that?**

We are reticent to focus on winter differences, as these are less well informed by the atmospheric $CO_2$ data and the absolute differences in NEE are relatively small.

**Line 441. I don't understand the line "TRENDY v8 data were downloaded from trendy-v8@trendy.ex.ac.uk.", since that is an email address, not a URL. Please provide a URL or DOI to a FAIR-aligned data archive where the data can be freely downloaded or, if the data is not available, then per this journal's data policy, a detailed explanation of why this is the case is required.**

The TRENDY data were downloaded and analyzed for this study, but these data were not generated by us and we do not control their data policy. This text has been re-worded to point to the TRENDY website:

[revised manuscript text omitted]

---

## Author Response (AR2)

Dear Dr. Bowling,

We appreciated the constructive comments of the reviewers. We have addressed the comments below. Reviewer/editor comments are shown in bold with our responses in blue. Line numbers refer to the tracked changes manuscript, and changes to the text are underlined.

**Anonymous referee #1**

**In general, I am satisfied with the author's thoughtful response to my comments (as well as author responses to reviewer #2 that expressed similar concerns) and I think that the paper is an important contribution to this issue of regional fluxes and their underlying processes. However, I am a bit confused about how the regional net fluxes are calculated in the revised figure 2. After all, the mismatch between the models and observationally constrained fluxes provides the rationale for the rest of the study. In the revised figure 2 there are apparent mismatches between OCO2 inversion estimates of NEE and the trendy model estimates of NEE, but not necessarily in the estimates of GPP. Therefore, I am surprised that these mismatches do not propagate to the inferred estimates of TER. For instance, over the cold region in July there is ~ 2 gC m-2 day-1 difference between IS NEE and TRENDY NEE and virtually no difference in GPP and yet the difference in TER seems to only be ~ 1 gC m-2 day-1. How is this possible? Were fire emissions included in the inversion estimates but probably not in the models, in which case we are not comparing the same fluxes? If we do not see big differences in TER, then the elegant thought experiment of Rh variability with soil depth seems less meaningful. I believe the authors that the DGVMs probably do a crummy job of simulating soil Rh, but if they are getting overall TER correct for the wrong reasons do we care that much.**

This may be a bit of an optical illusion as the magnitude of data-model mismatches are very similar for NEE and TER plots. One could be misled due to the fact that the y-axis shows a much larger ranges for GPP and TER than for NEE. To mitigate this, we have revised the Fig. 2 and 3 so that the background grid has horizontal lines every 0.5 gC m$^{-2}$ day$^{-1}$ for each plot (see below).

Both the TRENDY and inversion NEE estimates are consistent in that they do not include fire emissions. We describe that we subtract fire emissions from the inversions in Sec. 2.2.

[Figure]

Figure 2. Monthly carbon cycle fluxes (average of 2015, 2016 and 2018; 2017 is excluded due to an OCO-2 data gap). (a-c) Mean (solid line) and interquartile range (shaded area) of NEE for the ensemble of IS (red) and LNLG (blue) v9 OCO-2 MIP and for the TRENDY ensemble (green). (d-f) GPP for the TRENDY ensemble (green) and data-driven datasets (black). (g-i) TER simulated by the TRENDY ensemble (green) and calculated from combining the data-driven GPP with the IS (red) and LNLG (blue) v9 OCO-2 MIP NEE constraints.

[Figure]

Figure 3. Monthly carbon cycle fluxes (average of 2015, 2016 and 2018; 2017 is excluded due to an OCO-2 data gap). (a-c) Mean (solid line) and interquartile range (shaded area) of NEE for the ensemble of IS (red) and LNLG (blue) v9 OCO-2 MIP and for the TRENDY ensemble (green). (d-f) NPP for the TRENDY ensemble (green) and estimated from data-driven GPP (black). (g-i) $R_h$ simulated by the TRENDY ensemble (green) and calculated from combining the data-driven NPP with the IS (red) and LNLG (blue) v9 OCO-2 MIP NEE constraints. (j-l) Cumulative fraction of $R_h$ over the growing season. Figure S7 shows these fluxes per unit area

**Anonymous referee #2**

The manuscript is improved and I thank the authors for their careful edits. I think the authors have responded to all points that I had made, except for one, and so the manuscript is acceptable beyond the one caveat.

The one exception is the data availability statement regarding the TRENDY model output. The revised text reads, "TRENDY v8 gridded data can be accessed through the website https://sites.exeter.ac.uk/trendy." I believe that this is not correct, I can't find on that website exactly how to access the DGVM data, all that I can find under the "data policy" tab is that the TRENDY PIs need to be contacted for data. So I believe that this does not qualify as publicly accessible data. Biogeoscience's data availability policy https://www.biogeosciences.net/policies/data_policy.html states, "If the data are not publicly accessible, a detailed explanation of why this is the case is required." So please either provide a description of where and how the DGVM output data can be publicly accessed, or a detailed explanation of why it is not publicly accessible.

Yes, that is correct, the instructions from the site indicate to contact TRENDY PIs. We did this and the data was provided by Stephen Sitch. This has been clarified in the data availability statement:

"TRENDY v8 gridded data were accessed by contacting Stephen Sitch following the TRENDY data policy described on their website: https://sites.exeter.ac.uk/trendy."

I am not a member of TRENDY and only a data user. I have no ability to influence their data policy.